# Interchromosomal segmental duplication drives translocation and loss of *P. falciparum* histidine-rich protein 3

Nicholas J Hathaway[1†], Isaac E Kim[2,3†], Neeva WernsmanYoung[4], Sin Ting Hui[5], Rebecca Crudale[5], Emily Y Liang[5], Christian P Nixon[5], David Giesbrecht[5], Jonathan J Juliano[6,7,8], Jonathan B Parr[7,8], Jeffrey A Bailey[2,3,5]*

[1]Department of Medicine, University of Massachusetts Chan Medical School, Worcester, United States; [2]Center for Computational Molecular Biology, Brown University, Providence, United States; [3]Warren Alpert Medical School, Brown University, Providence, United States; [4]Department of Molecular Pharmacology, Physiology and Biotechnology, Brown University, Providence, United States; [5]Department of Pathology and Laboratory Medicine, Brown University, Providence, United States; [6]Department of Epidemiology, Gillings School of Global Public Health, University of North Carolina, Chapel Hill, United States; [7]Division of Infectious Diseases, Department of Medicine, School of Medicine, University of North Carolina, Chapel Hill, United States; [8]Curriculum in Genetics and Molecular Biology, School of Medicine, University of North Carolina at Chapel Hill, Chapel Hill, United States

*For correspondence:
jeffrey_bailey@brown.edu

†These authors contributed equally to this work

**Abstract** Most malaria rapid diagnostic tests (RDTs) detect *Plasmodium falciparum* histidine-rich protein 2 (PfHRP2) and PfHRP3, but deletions of *pfhrp2* and *phfrp3* genes make parasites undetectable by RDTs. We analyzed 19,313 public whole-genome-sequenced *P. falciparum* field samples to understand these deletions better. *Pfhrp2* deletion only occurred by chromosomal breakage with subsequent telomere healing. *Pfhrp3* deletions involved loss from *pfhrp3* to the telomere and showed three patterns: no other associated rearrangement with evidence of telomere healing at breakpoint (Asia; Pattern 13⁻TARE1); associated with duplication of a chromosome 5 segment containing multidrug-resistant-1 gene (Asia; Pattern 13⁻5$^{++}$); and most commonly, associated with duplication of a chromosome 11 segment (Americas/Africa; Pattern 13⁻11$^{++}$). We confirmed a 13–11 hybrid chromosome with long-read sequencing, consistent with a translocation product arising from recombination between large interchromosomal ribosome-containing segmental duplications. Within most 13⁻11$^{++}$ parasites, the duplicated chromosome 11 segments were identical. Across parasites, multiple distinct haplotype groupings were consistent with emergence due to clonal expansion of progeny from intrastrain meiotic recombination. Together, these observations suggest negative selection normally removes 13⁻11$^{++}$*pfhrp3 deletions*, and specific conditions are needed for their emergence and spread including low transmission, findings that can help refine surveillance strategies.

## eLife assessment

This work provides **important** insight into the mechanisms of hrp2 and particularly hrp3 deletion generation. The generation of additional long-read data alongside a new analysis of 19,000 public short-read sequenced genomes makes this the most detailed investigation currently available on this topic, which has high public health importance and also basic biological interest. The revised version of the manuscript provides **convincing** evidence for the proposed mechanisms.

## Introduction

### *P. falciparum* malaria remains a leading cause of childhood mortality in Africa

*Plasmodium falciparum* remains one of the most common causes of malaria and childhood mortality in Africa despite significant efforts to eradicate the disease (*WHO, 2022*). The latest report by the World Health Organization estimated 247 million cases of malaria and 619,000 fatalities in 2021 alone, with the vast majority of deaths occurring in Africa (*WHO, 2022*).

The mainstay of malaria diagnosis across Africa is no longer microscopy but rapid diagnostic tests (RDTs) due to their simplicity and speed. Their swift adoption, now totaling hundreds of millions a year, coupled with effective artemisinin-based combination therapies (ACTs) has led to significant progress in malaria control (*Chiodini, 2014*; *Poti et al., 2020*). The predominant and most sensitive falciparum malaria RDTs detect PfHRP2 and, to a lesser extent, its paralog PfHRP3 due to cross-reactivity.

### Increasing numbers of pfhrp2 and pfhrp3-deleted parasites escaping diagnosis by RDTs

Unfortunately, a growing number of studies have reported laboratory and field isolates with deletions of *pfhrp2* (PF3D7_0831800) and *pfhrp3* (PF3D7_1372200) in the subtelomeric regions of chromosomes 8 and 13, respectively. The resulting lack of these proteins allows the parasite to fully escape diagnosis by PfHRP2-based RDTs (*Baker et al., 2005*; *Cheng et al., 2014*; *Organization WH, 2011*; *Poti et al., 2020*; *Thomson et al., 2020*). *Pfhrp2/3*-deleted parasites appear to be spreading rapidly in some regions and have compromised existing test-and-treat programs, especially in the Horn of Africa (*Berhane et al., 2018*; *Feleke et al., 2021*; *Verma et al., 2018*; *WHO, 2020*). The prevalence of parasites with *pfhrp2* and *pfhrp3* deletion varies markedly across continents and regions in a manner not explained by RDT use alone. Parasites with these deletions are well-established in areas where PfHRP2-based RDTs have never been used routinely, such as parts of South America (*Thomson et al., 2020*). Studies in Ethiopia, where false-negative RDTs owing to *pfhrp2 and pfhrp3* deletions are common, suggest that the *pfhrp3* deletion arose first, given it is more prevalent and shows a shorter shared haplotype (*Feleke et al., 2021*). The reason why *pfhrp3* deletion occurred prior to *pfhrp2* remains unclear. A 1994 study of the HB3 laboratory strain reported frequent meiotic translocation of a *pfhrp3* deletion from chromosomes 13 to 11 (*Hinterberg et al., 1994*). An explanation of this mechanism, whether it might occur in natural populations, and how it relates to the initial loss of *pfhrp3* has not been fully explored.

### Precise pfhrp2 and pfhrp3 deletion mechanisms remain unknown

Studies of *P. falciparum* structural rearrangements are challenging, and *pfhrp2* and *pfhrp3* deletions are particularly difficult due to their position in complex subtelomeric regions. Subtelomeric regions represent roughly 5% of the genome, are unstable, and contain rapidly diversifying gene families (e.g. *var*, *rifin*, *stevor*) that undergo frequent conversion between chromosomes mediated by non-allelic homologous recombination (NAHR) and double-stranded breakage (DSB) and telomere healing (*Calhoun et al., 2017*; *Gardner et al., 2002*; *Lee et al., 2014*; *Mattei and Scherf, 1994*; *Reed et al., 2021*; *Vernick and McCutchan, 1988*; *Zhang et al., 2019*). Subtelomeric exchange importantly allows for unbalanced progeny without the usual deleterious ramifications of altering a larger proportion of a chromosome. Notably, newly formed duplications predispose to further duplications or other rearrangements through NAHR between highly identical paralogous regions. Together, this potentiates the rapid expansion of gene families and their spread across subtelomeric regions (*Conrad et al., 2010*; *Didelot et al., 2012*; *Kidd et al., 2010*; *Korbel et al., 2007*; *Mills et al., 2011*; *Parks et al., 2015*). The duplicative transposition of a subtelomeric region of one chromosome onto another chromosome frequently occurs in *P. falciparum*. Specifically, prior studies have found duplicative transposition events involving several genes, including *var2csa* and *cytochrome b* (*Bopp et al., 2013*; *Claessens et al., 2014*; *Sander et al., 2014*; *Sander et al., 2009*; *Zhang et al., 2019*). Notably, *pfhrp2* and *pfhrp3* are adjacent to but not considered part of the subtelomeric regions, and recombination of *var* genes does not result in the deletion of *pfhrp2* and *pfhrp3* (*Feleke et al., 2021*; *Otto et al., 2018a*).

Telomere healing, de novo telomere addition via telomerase activity, is associated with subtelomeric deletion events in *P. falciparum* that involve chromosomal breakage and loss of all downstream genes. Healing serves to stabilize the end of the chromosome. Deletion of the *P. falciparum* knob-associated histidine-rich protein (*KAHRP* or *pfhrp1*) and *pfhrp2* genes via this mechanism was first reported to occur in laboratory isolates (**Pologe and Ravetch, 1988**). Since then, studies have defined the critical role of telomerase in *P. falciparum* and additional occurrences affecting several genes, including *pfhrp1*, *Pf332*, and *Pf87* in laboratory isolates (**Bottius et al., 1998**; **Mattei and Scherf, 1994**; **Scherf and Mattei, 1992**). For *pfhrp1* and *pfhrp2,* this deletion mechanism only occurred in laboratory isolates but not in clinical samples, suggesting the genes have important roles in normal infections, and their loss is selected against (**Scherf and Mattei, 1992**).

An improved understanding of the patterns and mechanisms of *pfhrp2* and *pfhrp3* deletions can provide important insights into how frequently they occur and the evolutionary pressures driving their emergence and help inform control strategies. We examined the pattern and nature of *pfhrp2* and *pfhrp3* deletions using available whole-genome sequences and additional long-read sequencing. The objectives of this study were to determine the *pfhrp3* deletion patterns along with their geographical associations and sequence and assemble the chromosomes containing the deletions using long-read sequencing. Our findings shed light on geographical differences in *pfhrp3* deletion patterns, their mechanisms, and how they likely emerged, providing key information for improved surveillance.

## Results

### Pfhrp2 and pfhrp3 deletions in the global *P. falciparum* genomic dataset

We examined all publicly available Illumina whole-genome sequencing (WGS) data from global *P. falciparum* isolates as of January 2023, comprising 19,313 field samples and lab isolates (**Supplementary file 1**). We analyzed the genomic regions containing *pfhrp2* on chromosome 8 and *pfhrp3* on chromosome 13 to detect nucleotide and copy number variation (e.g. deletions and duplications) using local haplotype assembly and sequencing depth. Regions on chromosomes 5 and 11 associated with these duplicates were also analyzed based on associations with patterns seen within chromosomes 8 and 13 (**Supplementary file 2**, **Supplementary file 3**, Methods). Samples were eliminated for downstream analysis if they had less than 5 x median genomic coverage and selective whole genome amplification (sWGA) samples with significant variability on visible inspection. This resulted in 3321 samples being eliminated, leaving 15,992 samples for further analysis. We removed six samples that showed signs of deletions but also had a complexity of infection (COI) greater than 1 to eliminate any artifact introduced by mixed infections from downstream analysis. We identified 26 parasites with *pfhrp2* deletion and 168 with *pfhrp3* deletion. Twenty field samples contained both deletions; 11 were found in Ethiopia, six in Peru, and three in Brazil, and all had the $13^{-}11^{++}$ *pfhrp3* deletion pattern. Across all regions, *pfhrp3* deletions were more common than *pfhrp2* deletions; specifically, *pfhrp3* deletions and *pfhrp2* deletions were present in Africa in 42 and 11, Asia in 51 and 3, and South America in 75 and 11 parasites. It should be noted that these numbers are not accurate measures of prevalence, given that the publicly available WGS specimens utilized in this analysis come from locations and time periods that commonly used RDT positivity for collection. Furthermore, subtelomeric regions are difficult to sequence and assemble, meaning these numbers may be significantly underestimated.

### Pfhrp2 deletion associated with variable breakpoints and telomeric healing

We further examined the breakpoints of 26 parasites (24 patient parasites and two lab isolates) (**Figure 1—figure supplement 1**). Twelve parasites showed evidence of breakage and telomeric healing as suggested by telomere-associated tandem repeat 1 (TARE1) (**Vernick and McCutchan, 1988**) sequence contiguous with the genomic sequence at locations where coverage drops to zero (**Figure 1—figure supplement 2**). Most breakpoints occur within *pfhrp2*, found in 9 South American parasites and lab isolate D10 (**Figure 1—figure supplement 2**). The other *pfhrp2*-deleted parasites did not have detectable TARE1 or evidence of genomic rearrangement but were amplified with sWGA, limiting the ability to detect the TARE1 sequence. Thus, *pfhrp2* deletion likely occurs solely through breakage events with subsequent telomeric healing.

## Three distinct pfhrp3 deletion patterns with geographical associations

Exploration of read depth revealed three distinct deletion copy number patterns associated with *pfhrp3* deletion (chromosome 13: 2,840,236–2,842,840): first, sole deletion of chromosome 13 starting at various locations centromeric to *pfhrp3* to the end of the chromosome with detectable TARE1 telomere healing and unassociated with other rearrangements (**pattern 13-TARE1**); second, deletion of chromosome 13 from position 2,835,587 to the end of the chromosome and associated with duplication of a chromosome 5 segment from position 952,668–979,203,, which includes *pfmdr1* (**pattern 13-5++**); and third, deletion of chromosome 13 commencing just centromeric to *pfhrp3* and extending to the end of the chromosome and associated with duplication of the chromosome 11 subtelomeric region (**pattern 13-11++**) (*Figure 1*). Among the 168 parasites with *pfhrp3* deletion, 20 (11.9%) were pattern **13-TARE1**, 28 (16.6%) were pattern **13-5++**, and the majority with 120 (71.4%) demonstrated pattern **13-11++**. Pattern 13-11++ was almost exclusively found in parasites from Africa (n=39) and the Americas (n=75), with six in Asia, while 13-5++ was only observed in Asia (*Figure 1*). As a result of our short-read analysis demonstrating these three patterns and discordant reads between the chromosomes involved, chromosomes 5, 11, and 13 were further examined. No other chromosomes had associated discordant reads or changes in read coverage.

## Pattern 13-TARE1 associated with telomere healing

The 20 parasites with pattern 13-TARE1 with no association with any other chromosome rearrangement had deletions of the core genome averaging 19 kb (range: 11–31 kb). Of these 13-TARE1 deletions, 19 out of 20 had detectable TARE1 (**pattern 13-TARE**) adjacent to the breakpoint, consistent with telomere healing (*Figure 1—figure supplement 3*).

## Pattern 13-5++ associated with NAHR-mediated pfmdr1 duplication and subsequent telomere healing

The 28 parasites with deletion pattern 13-5++ had a consistent loss of 17.9 kb of chromosome 13 and a gain of 25 kb from chromosome 5. These isolates have evidence of a genomic rearrangement that involves a 26 bp AT di-nucleotide repeat at 2,835,587 on chromosome 13, and a 20 bp AT di-nucleotide repeat at 979,203 on chromosome 5. Analysis revealed paired-end reads with discordant mapping, with one read mapping to chromosome 13 and the other to chromosome 5. Reads assembled from these regions form a contig of a unique sequence that connects chromosome 13 (position 2,835,587) to chromosome 5 (position 979,203) in reverse orientation. Read depth coverage analysis revealed more than a twofold increase on chromosome 5 from 979,203–952,668,, with the TARE1 sequence contiguously extending from 952,668, consistent with telomere healing. This 25 kb duplication contained several genes, including intact PF3D7_0523000 (*pfmdr1*) (*Figure 1—figure supplement 4* and *Figure 1—figure supplement 5*), and TARE1 transition occurred within the gene PF3D7_0522900 (a zinc finger gene). Further read depth, discordant read, and assembly analysis revealed four 13-5++ parasites that, in addition to the chromosome 5 segment duplication on chromosome 13, had the described *pfmdr1* tandem duplication on chromosome 5 associated with drug resistance, resulting in over threefold read depth across *pfmdr1 gene* (*Figure 1—figure supplement 4*; *Nair et al., 2007*). The parasites containing this translocation do not appear to be a clonal expansion based on whole genome IBD (*Figure 1—figure supplement 6*, *Figure 1—figure supplement 7*), which suggests this event occurred multiple times with the same TARE1 healing point and chromosome 13/5 transition point or that it occurred once with this hybrid chromosome 13–5 being passed down to progeny during sexual recombination and maintained in further progeny thereafter.

## Pattern 13-11++ predominated in the Americas and Africa

Pattern 13-11++ was observed in 75 American parasites, 39 African parasites, and 6 Asian parasites (*Figure 1*). Of the 120 parasites with this *pfhrp3* deletion pattern, 96 (73 of the 75 American, 18 of 39 African parasites, and 5 of 6 Asian) had near-identical copies of the chromosome 11 duplicated region. Near-identical copies were defined as having ≥99% identity (same variant microhaplotype between copies) across 382 variant microhaplotypes (*Supplementary file 3*) within the duplicated region, far less than normal between parasite allelic differences. These 96 parasites containing identical copies did not all share the same overall haplotype but showed nine major haplotypes found in more than one sample and 14 haplotypes found only in one sample each (*Figure 2*, *Figure 2—figure*

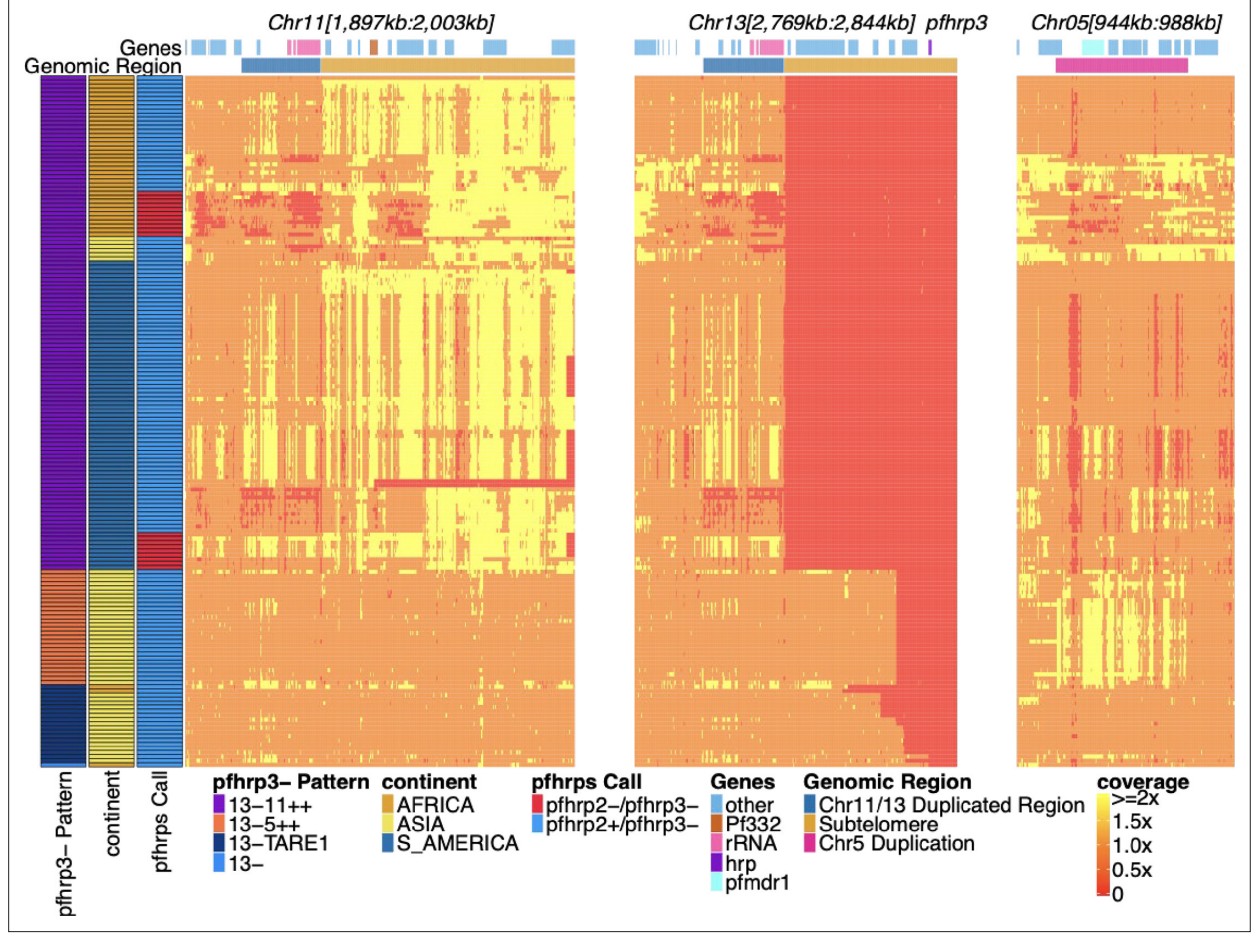

**Figure 1.** Pfhrp2/3 deleted parasites with altered sequence coverage across regions of chromosomes 11, 13, and 5. Sequence coverage heatmap of *pfhrp3* deletion associated regions of chromosomes 11 (1,897,151–2,003,328 bp), 13 (2,769,916–2,844,785 bp), and 5 (944,389–988,747 bp) in the 168 samples with evidence of *pfhrp3* deletion out of the 19,313 publicly available samples. The regions from chromosomes 11 and 13 are at the end of their core region, while the region from chromosome 5 is the region around *pfmdr1* involved in its duplication event. Each row is a whole-genome sequencing (WGS) sample, and each column is normalized coverage. The top annotation along chromosomes depicts the location of genes with relevant genes colored: rRNA (light pink), *pf332* (red-orange), *pfhrp3* (purple), *pfmdr1* (electric-blue), and all other genes are colored light-blue. The second row delineates significant genomic regions: The chromosome 11/13 duplicated region (dark blue), the subtelomere regions of chr11/13 (orange), and the chromosome 5 duplicated region (fuchsia). The left annotation for samples includes the genomic rearrangement/deletion pattern (patterns with telomere-associated tandem repeat 1 (TARE1) have evidence of TARE1 addition following deletion), the continent of origin, and *pfhrp2/3* deletion calls. Increased variation and biases in coverage correlate with *P. falciparum* selective whole-genome amplification (sWGA), which adds variance and biases to the sequence coverage before sequencing.

The online version of this article includes the following figure supplement(s) for figure 1:

**Figure supplement 1.** Genome coverage of isolates with evidence of *pfhrp2* deletion.

**Figure supplement 2.** Coverage of sub-telomeric region of chromosome 8 before *pfhrp2* of parasites with *pfhrp2* deletion.

**Figure supplement 3.** Coverage of chromosome 13 for parasites with *pfhrp3* deletion pattern 13⁻TARE1.

**Figure supplement 4.** Coverage of chromosome 5 for parasites with *pfhrp3* deletion pattern 13⁻TARE1 and 13⁻5⁺⁺.

**Figure supplement 5.** Chromosome 5 duplicated region microhaplotypes.

**Figure supplement 6.** Whole genome identity by descent (IBD) between all parasites with a genomic deletion.

**Figure supplement 7.** Whole genome identity by descent (IBD) between *pfhrp3* deletion pattern 13⁻5⁺⁺ parasites.

*supplement 1*). The remaining 22 parasites varied within this region; on average, 10.2% of variant sites differed between the copies (min 83.8% identity). Of these, three haplotypes were found in more than one sample (*Figure 2*, *Figure 2—figure supplement 2*). The different haplotypes showed geographical separation, with distinct haplotypes observed in American and African strains (*Figure 2—figure supplement 3*, *Figure 2—figure supplement 4*, *Figure 2—figure supplement 5*, *Figure 2—figure*

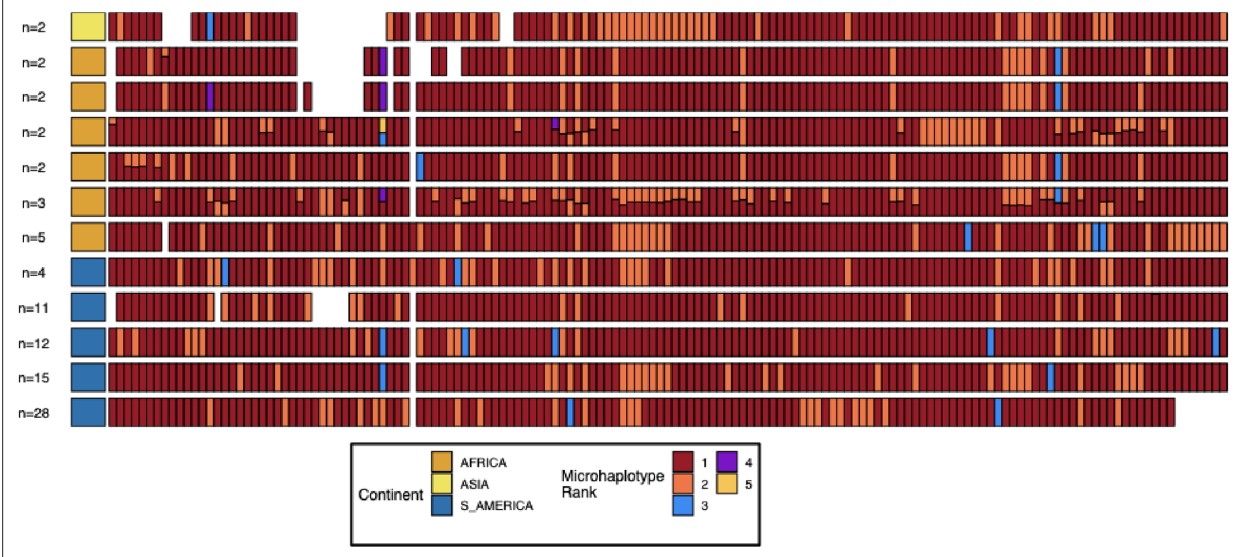

**Figure 2.** Microhaplotype patterns for the duplicated portion of chromosome 11 in 13⁻11++ parasites form 11 distinct haplotype groups with a geographic distinction between Africa and the Americas. Each row represents a group of 13⁻11++ parasites based on shared haplotypes on the chromosome 11 duplicated segment. The number of parasites and continent of origin are on the left for each group. Each column is a different genomic region across the duplicated portion of chromosome 11. In each column, the microhaplotype is colored by the prevalence of each microhaplotype (named Microhaplotype Rank), with 1=red being the most prevalent, 2=orange being the second most prevalent, and so forth. If more than one microhaplotype for a parasite is present at a genomic location, its height is relative to within-parasite frequency. Only sites with microhaplotype variation are shown (n=202). Most parasites show singular haplotypes at variant positions despite two copies consistent with identical haplotypes in the group, and when there are multiple microhaplotypes, the relative frequencies are 50/50, consistent with two divergent copies. Overall, haplotype groups are markedly different, consistent with separate translocations emerging and spreading independently.

The online version of this article includes the following figure supplement(s) for figure 2:

**Figure supplement 1.** Chromosome 11 Duplicated Segment *pfhrp3* deletion Pattern 13⁻11++ parasites with perfect copies.

**Figure supplement 2.** Chromosome 11 Duplicated Segment *pfhrp3* deletion Pattern 13⁻11++ parasites with divergent chromosome 11 copies.

**Figure supplement 3.** Jaccard similarity between parasites for chromosome 11 duplicated segment for *pfhrp3* deletion pattern 13⁻11++ parasites.

**Figure supplement 4.** Chromosome 11 Duplicated Segment *pfhrp3* deletion Pattern 13⁻11++ parasites.

**Figure supplement 5.** Jaccard similarity for chromosome 11 duplicated segment.

**Figure supplement 6.** Chromosome 11 duplicated segment coverage for *pfhrp3* deletion Pattern 13⁻11++ parasites SD01, HB3, and Salvador 1.

*supplement 2*). The haplotypes for the segment of chromosome 11 found within 13⁻11++ parasites could also be found within the parasites lacking the 13⁻11++ translocation (*Figure 2—figure supplement 5*). The lab isolates with 13⁻11++ SD01 (Africa) and Santa-Luca-Salvador-1 (America) have perfect copies, while HB3 (America) has divergent copies (*Figure 2—figure supplement 6*).

## Pattern 13⁻11++ breakpoint occurs in a segmental duplication of ribosomal genes on chromosomes 11 and 13

Pattern 13⁻11++ has a centromeric breakpoint consistently occurring within a 15.2 kb interchromosomal segmental duplication on chromosomes 11 and 13 (Chr11-1918028-1933288 (bp = 15,274), Chr13-2792021-2807295 (bp = 15,260)). It was the largest duplication in the core genome based on an all-by-all unique k-mer genome comparison using nucmer (*Kurtz et al., 2004*; *Supplementary file 5*). The two copies on chromosome 11 and chromosome 13 in the reference genome were 99.0% identical (*Figure 3*) and oriented similarly. Each copy contained a centromeric 7 kb region encoding 2–3 undefined protein products (98.9% identity) and a telomeric 8 kb nearly identical region (99.7% identity) containing one of the two S-type (*Gardner et al., 2002*) ribosomal genes (S=sporozoite), which are primary expressed during life cycle stages in mosquito vector (*Figure 3*). Pairwise alignment of the chromosomes 11 and 13 paralogs showed similar allelic and paralogous identity levels. No consistent nucleotide differences were found between the paralogs, leading to no distinct separation between copies when clustered (*Figure 3*). This suggests ongoing interchromosomal exchanges or

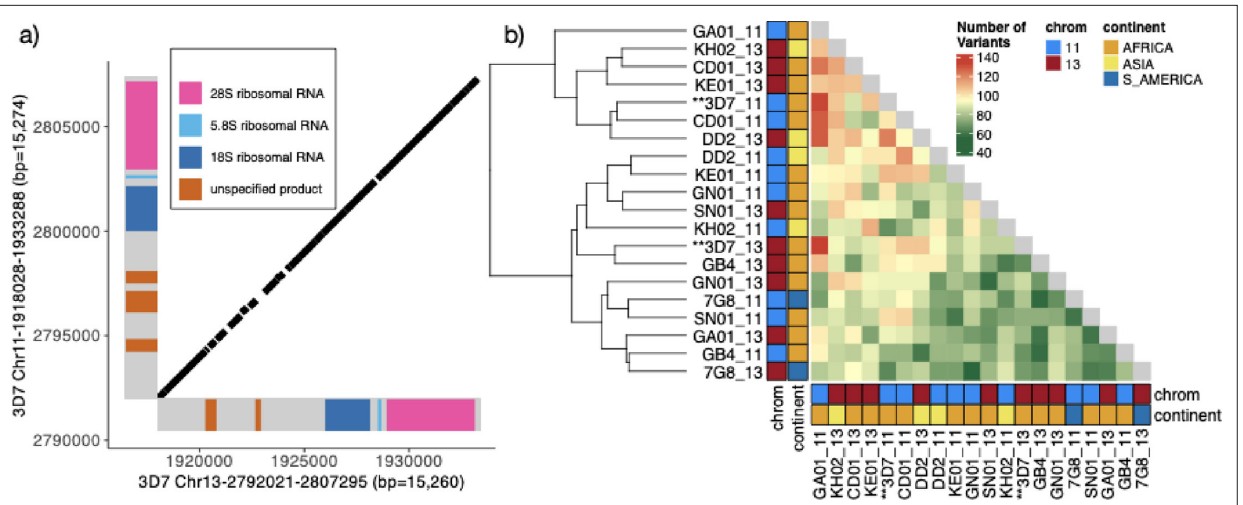

**Figure 3.** Characterization of the 15.2 kb segmental duplication containing ribosomal genes on chromosomes 11 and 13. (**a**) Alignment of 3D7 reference genome copies on chromosome 11 (1,918,028–1,933,288 bp) and chromosome 13 (2,792,021–2,807,295 bp). These two regions are 99.3% identical. The diagonal black bars show 100% conserved regions of at least 30 bp in length, representing 89.1% of the alignment. Gene annotation is colored. (**b**) Comparison by pairwise alignments of the duplicated copies from non-*pfhrp3* deleted strains (**Otto et al., 2018a**) assemblies (n=10) does not show a discrete separation of the paralogs with copies intermixed (chromosome 11 in blue and 13 in red). All copies are ≥99.0% similar, with the number of variants between segments ranging from 55 to 133 with no clear separation by continent or chromosome.

The online version of this article includes the following figure supplement(s) for figure 3:

**Figure supplement 1.** Jaccard similarity for Chromosome 11/13 15.2 kb duplicated region for *pfhrp3* deletion pattern 13[-]11[++] parasites.

**Figure supplement 2.** Chromosome 11/13 15.2 kb duplicated region for *pfhrp3* deletion pattern 13[-]11[++] parasites.

**Figure supplement 3.** Gene Annotations of Chromosome 8 of PacBio-assembled *P. Laverania* Genomes.

**Figure supplement 4.** Gene Annotations of Chromosome 11 of PacBio-assembled *P. Laverania* Genomes.

**Figure supplement 5.** Gene Annotations of Chromosome 13 of PacBio-assembled *P. Laverania* Genomes.

**Figure supplement 6.** Gene annotations of chromosome 11 of PacBio-assembled genomes.

**Figure supplement 7.** Gene annotations of chromosome 13 of PacBio-assembled genomes.

conversion events maintaining paralog homogeneity. There is less separation by continent in this region when clustering by the variation in this region compared to the duplicated chromosome 11 segment (*Figure 3—figure supplement 1*, *Figure 3—figure supplement 2*).

## Ribosomal gene segmental duplication exists in closely related *P. praefalciparum*

To look at the conservation of the segmental duplication containing the ribosomal genes, we examined genomes of closely related *Plasmodium* parasites in the *Laverania* subgenus, which comprised *P. falciparum* and other *Plasmodium* spp found in African apes. The *Plasmodium praefalciparum* genome, which is *P. falciparum*'s closest relative, having diverged about 50,000 years ago (*Otto et al., 2018b*), also contained similar S-type rRNA loci on chromosomes 11 and 13 and had a similar gene synteny to *P. falciparum* in these regions and the region on chromosome 8 neighboring *pfhrp2* (*Figure 3—figure supplement 3*, *Figure 3—figure supplement 4*, and *Figure 3—figure supplement 5*). *P. praefalciparum* contained the 15.2 kb duplicated region on both chromosomes 11 and 13 and was 96.7% similar to the 3D7 duplicated region. Other *Laverania* genomes (*Otto et al., 2018b*) were not fully assembled within their subtelomeric regions.

## Previous PacBio assemblies did not fully resolve chromosome 11 and 13 subtelomeres

Given pattern 13[-]11[++] was suggestive of duplication-mediated recombination leading to translocation, we examined high-quality PacBio genome assemblies of SD01 from Sudan and HB3 from Honduras, both containing the *pfhrp3* deletion. However, the Companion (*Steinbiss et al., 2016*)

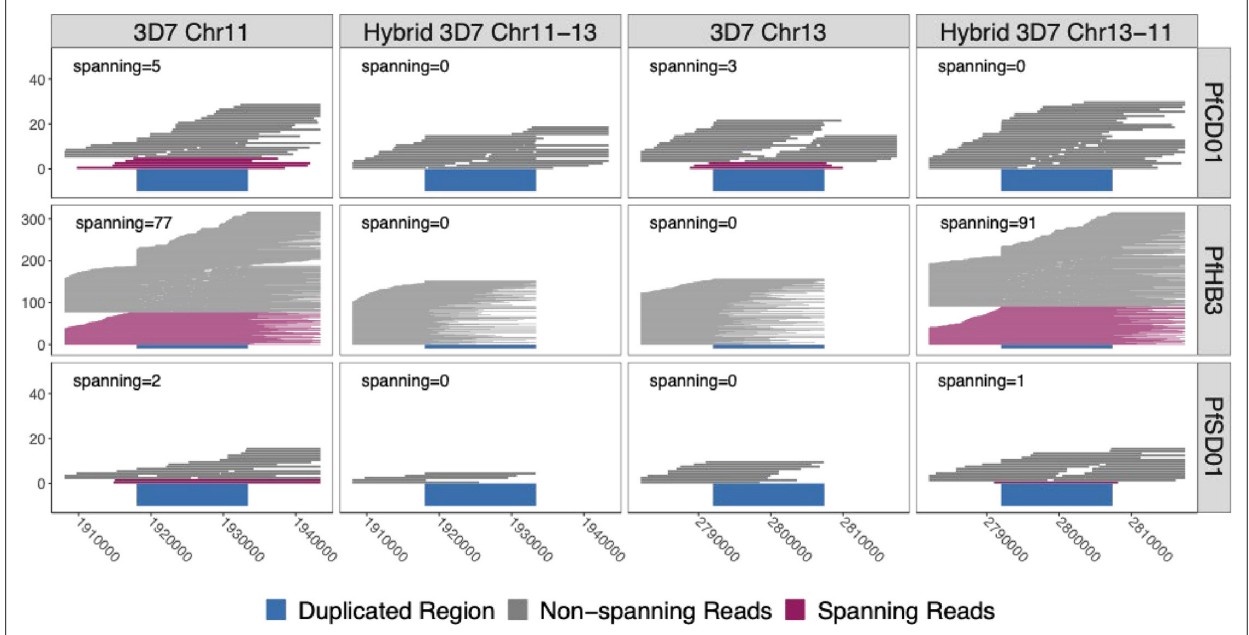

**Figure 4.** Long reads spanning the 15 kb duplicated region confirm the presence of translocated chromosome 13–11 in *pfhrp3*-deleted HB3 (Americas) and SD01 (Africa) but not *pfhrp3*-intact chromosomes. PacBio and Nanopore read >15 kb for HB3, SD01, and CD01 are shown aligned to normal chromosomes 11 and 13 and hybrid chromosomes 11–13 and 13–11 constructed from the 3D7 sequence. Reads that completely span the segmental duplication (dark blue) anchoring in the unique flanking sequence are shown in maroon. Spanning reads are mapped only to this one location, whereas the non-spanning reads are mapped to both the hybrid or normal chromosomes as these chromosome segments are identical. SD01 and HB3 only have reads that span the duplicated region on chromosome 11, but no reads that span out of the duplicated region on chromosome 13. Instead, SD01 and HB3 have spanning reads across the hybrid chromosome 13–11. Other non-deleted isolates had spanning reads mapped solely to normal chromosomes, exemplified by CD01 (top row). No isolates had spanning reads across the hybrid 11–13 chromosome.

The online version of this article includes the following figure supplement(s) for figure 4:

**Figure supplement 1.** Chromosome 11/13 15.2 kb duplicated region for parasites SD01, HB3, and Salvador 1.

**Figure supplement 2.** Spanning PacBio and Nanopore Reads across the duplicated region for SD01.

gene annotations of chromosome 11 (*Otto et al., 2018a*) showed that these strains were not fully assembled in the relevant regions (*Figure 3—figure supplement 6* and *Figure 3—figure supplement 7*).

## Combined analysis of additional Nanopore and PacBio reads confirmed a segmental duplicated region of the normal chromosome 11 and hybrid chromosome 13-11

To better examine the genome structure of pattern 13-11++, we whole-genome sequenced the 13-11++ isolates HB3 and SD01 with long-read Nanopore technology. We generated 7350 Mb and 645 Mb of data, representing an average coverage of 319 x and 29.3 x for HB3 and SD01, respectively. We combined our Nanopore data with the publicly available PacBio sequencing data and tested for the presence of hybrid chromosomes using a two-pronged approach: (1) mapping the long reads directly to normal and hybrid chromosome 11/13 constructs and (2) optimized de-novo assembly of the higher quality Nanopore long reads.

To directly map reads, we constructed 3D7-based representations of hybrid chromosomes 13–11 and 11–13 by joining the 3D7 chromosomal sequences at breakpoints in the middle of the segmental duplication (**Methods**). We then used minimap2 (*Li, 2018*), width default settings, to align all PacBio and Nanopore reads for each isolate to the normal and hybrid constructs to detect reads completely spanning the duplicated region, extending at least 50 bp into flanking unique regions (*Figure 4*). HB3 had 77 spanning reads across normal chromosome 11 and 91 spanning reads across hybrid chromosome 13–11. SD01 had two chromosome 11 spanning reads and one 13–11 chromosome spanning read. SD01 had few spanning reads due to lower overall Nanopore reads secondary to

insufficient input sample. Further analysis of SD01 revealed four regions within this duplicated region with chromosome 11 and 13-specific nucleotide variation, which was leveraged to further bridge this region for additional confirmation given SD01's low coverage (*Figure 4—figure supplement 1* and *Figure 4—figure supplement 2*). Neither isolate had long reads spanning normal chromosome 13 or hybrid 11–13, representing the reciprocal translocation product (*Figure 4*). Importantly, the other 12 isolates with intact *pfhrp3* from the PacBio dataset (*Otto et al., 2018a*) all had reads consistent with normal chromosomes -- reads spanning chromosome 11 and chromosome 13 and no reads spanning the hybrid 13–11 or 11–13 chromosomes (*Figure 4*). Thus, long reads for HB3 and SD01 confirmed the presence of a hybrid 13–11 chromosome.

## De novo long-read assemblies of pfhrp3-deleted strains further confirmed hybrid 13-11 chromosome

To further examine the parasites with hybrid 13–11 chromosomes and exclude potentially more complicated structural alterations involving other genome regions, de novo whole-genome assemblies were created for the HB3 and SD01 lab strains from Nanopore long reads. Assembly statistics were generated using the program quast (*Gurevich et al., 2013*) and the 3D7 reference genome. HB3 assembly yielded 16 contigs representing complete chromosomes (N50 1,5985,898 and L50 5).

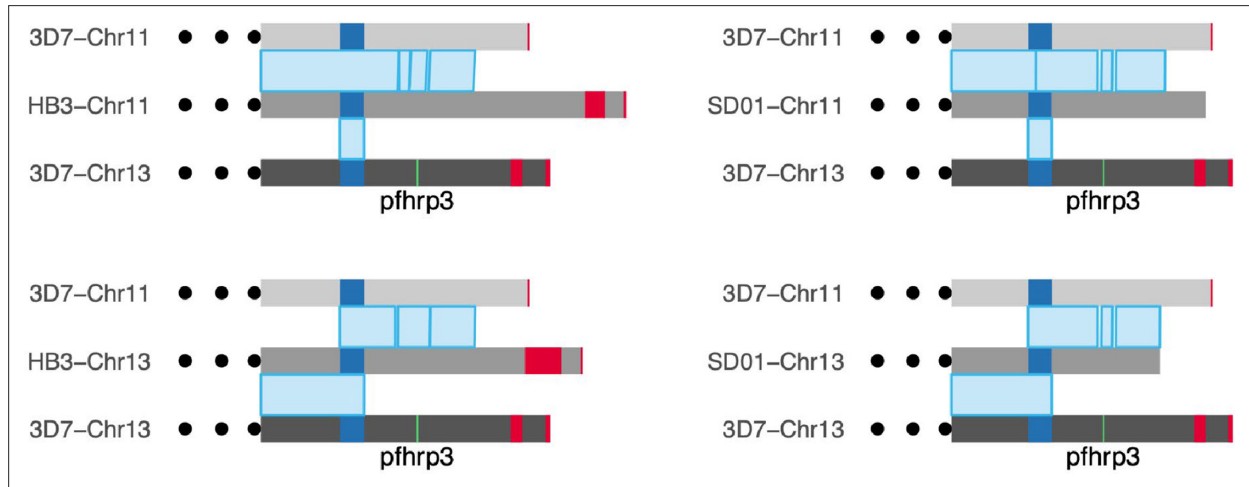

**Figure 5.** A comparison of long-read assemblies of chromosomes 11 and 13 of HB3 and SD01 with the reference genome 3D7 confirms hybridized chromosomes 13–11. On top, chromosome 11 of HB3 and SD01 mapped entirely to the reference chromosome 11 of 3D7, with the segmental duplication region in dark blue mapped to both 11 and 13. The assembly of chromosome 13 of HB3 and SD01 maps to the reference chromosome 13 of 3D7 up through the segmentally duplicated region, but after the duplication (where *pfhrp3* (green) should be found), it maps to chromosome 11 of 3D7 instead of chromosome 13. Red blocks mark telomere-associated repetitive elements (TARE) sequence. Displaying only from 50 kb upstream from the duplicated region to the end of the chromosomes. Chromosome 11 on 3D7 spans 1,918,029–2,038,340 (120,311bp in length) and chromosome 13 on 3D7 spans 2,792,022–2,925,236 (133,214bp in length).

The online version of this article includes the following figure supplement(s) for figure 5:

**Figure supplement 1.** Exact Matches between nanopore-assembled HB3 chromosome 13 with HB3 chromosome 11, 3D7 chromosomes 11, 13.

**Figure supplement 2.** Annotation of HB3 chromosomes 11 and 13–11.

**Figure supplement 3.** Annotation of SD01 chromosomes 11 and 13.

**Figure supplement 4.** Chromosome 11/13 15.2 kb duplicated region for *pfhrp3* deletion pattern 13⁻11⁺⁺ parasites with identical chromosome 11 segment haplotypes.

**Figure supplement 5.** Whole genome IBD between *pfhrp3* deletion pattern 13⁻11⁺⁺ parasites.

**Figure supplement 6.** Genome coverage chromosome 8, 11, and 13 regions of isolates with subtelomere deletion of chromosome 11.

**Figure supplement 7.** Windows of interest chromosomes 8, 11, 13.

**Figure supplement 8.** Windows of interest chromosome 05 around *pfmdr1*.

**Figure supplement 9.** Chromosome 11 Duplicated Segment *pfhrp3* deletion pattern 13⁻11⁺⁺ parasites biallelic single nucleotide polymorphisms (SNPs).

**Figure supplement 10.** Jaccard similarity using biallelic single nucleotide polymorphisms (SNPs) between parasites for chromosome 11 duplicated segment for *pfhrp3* deletion pattern 13⁻11⁺⁺ parasites.

TARE-1 (*Vernick and McCutchan, 1988*) was detected on the ends of all chromosomes except for the 3' end of chromosome 7 and the 5' end of chromosome 5, indicating that telomere-to-telomere coverage had been achieved. SD01, with lower sequencing coverage, had a more disjointed assembly with 200 final contigs (N50 263,459 and L50 30). The HB3 and SD01 assemblies both had a chromosome 11 that closely matched normal 3D7 chromosome 11 and a separate hybrid 13–11 that closely matched 3D7 chromosome 13 until the ribosomal duplication region, where it then subsequently best-matched chromosome 11 (*Figure 5*, *Figure 5—figure supplement 1*). HB3's 11 and hybrid 13–11 chromosomes had TARE-1 at their ends (*Vernick and McCutchan, 1988*), indicating that these chromosomes were complete. These new assemblies were further annotated for genes by Companion (*Steinbiss et al., 2016*). The contig matching the hybrid 13–11 for both strains essentially contained a duplicated portion of chromosome 11 telomeric to the ribosomal duplication. The duplicated genes within this segment included *pf332* (PF3D7_1149000), two ring erythrocyte surface antigens genes (PF3D7_1149200, PF3D7_1149500), three PHISTs genes, a FIKK family gene, and two hypothetical proteins, and ends with a DnaJ gene (PF3D7_1149600) corresponding to 3D7 genes PF3D7_1148700 through PF3D7_1149600 (*Figure 5—figure supplement 2* and *Figure 5—figure supplement 3*). Homology between HB3 chromosomes 11 and 13–11 continued up through a *rifin*, then a *stevor* gene, and then the sequence completely diverged in the most telomeric region with a different gene family organization structure, but both consisting of *stevor*, *rifin*, and *var* gene families along with other paralogous gene families (*Figure 5—figure supplement 2*). The chromosome 13–11 SD01 contig reached the DNAJ protein (PF3D7_1149600) and terminated (*Figure 5—figure supplement 3*), while normal 11 continued through two *var* genes and four rifin genes, likely because the assembly was unable to contend with the near complete identical sequence between the two chromosomes. Examination of the longer normal chromosome 11 portion revealed twofold coverage and no variation. Therefore, SD01 likely has identical chromosome 11 segments intact to the telomere of each chromosome.

Analysis of the 11 other PacBio assemblies (*Otto et al., 2018a*) with normal chromosome 11 showed that homology between strains also ended at this DnaJ gene (PF3D7_1149600) with the genes immediately following being within the *stevor*, *rifin*, and *var* gene families among other paralogous gene families. The genes on chromosome 13 deleted in the hybrid chromosome 13–11 corresponded to 3D7 genes PF3D7_1371500 through PF3D7_1373500 and include notably *pfhrp3* and *EBL-1* (PF3D7_1371600). The de-novo long-read assemblies of HB3 and SD01 further confirmed the presence of a normal chromosome 11 and hybrid chromosome 13–11 without other structural alterations.

## Genomic refinement of breakpoint location for 13$^-$11$^{++}$

To better define the breakpoint, we examined microhaplotypes within the 15.2 kb ribosomal duplication for the 98 13$^-$11$^{++}$ parasites containing near-perfect chromosome 11 segments (*Figure 2—figure supplement 1*). Within each parasite, the microhaplotypes in the telomeric region are identical, consistent with a continuation of the adjacent chromosome 11 duplication. However, for nearly all parasites, as the region traverses towards the centromere within the ribosomal duplication, there is an abrupt transition where the haplotypes begin to differ. These transition points vary but are shared within specific groupings correlating with the chromosome 11 microhaplotype clusters (*Figure 5—figure supplement 4*). These transition points likely represent NAHR exchange breakpoints, and their varied locations further support that multiple intrastrain translocation events have given rise to 13$^-$11$^{++}$ parasites in the population. When comparing the whole genome fraction of IBD sites between these parasites, there are distinctive haplotype groups within the chromosome 11 haplotype clusters within the biggest clusters 01 (n=28) and 03 (n=12) (*Figure 5—figure supplement 5*). The 01 cluster has three distinct groups, 03 cluster has five distinct groups. These groups are consistent with the distinct differences in the 15.2 kb duplicated region (see *Figure 5—figure supplement 4*), which would be consistent with different translocation events creating the same duplication segment of chromosome 11.

## Discussion

Here, we used publicly available Illumina short-read and PacBio long-read from parasites across the world and newly generated Nanopore long-read sequencing data to identify *pfhrp2* and *pfhrp3* deletions and their mechanisms in field *P. falciparum* parasites. The limited number of *pfhrp2*-deleted strains showed chromosome 8 breakpoints predominantly in the gene with evidence of telomere healing, a common repair mechanism in *P. falciparum* (*Calhoun et al., 2017*; *Lee et al., 2014*). We found that *pfhrp3* deletions occurred through three different mechanisms. The least common mechanism involved simple breakage loss of chromosome 13 from *pfhrp3* to the telomere, followed by telomere healing (13$^-$TARE1 pattern). The second most common pattern 13$^-$5$^{++}$ was likely the result of NAHR, within 20–28 bp di-nucleotide AT repeats translocating a 26,535 bp region of chromosome 5 containing *pfmdr1* onto chromosome 13, thereby duplicating *pfmdr1* and deleting *pfhrp3*. There appeared to be one origin of 13$^-$5$^{++}$, which was only observed in the Asia population, and its continued presence was potentially driven by the added benefit of *pfmdr1* duplication in the presence of mefloquine. The most common pattern, 13$^-$11$^{++}$, predominated in the Americas and Africa and was the result of NAHR between chromosome 11 and 13 within the large 15.2 kb highly-identical ribosomal duplication, translocating and thereby duplicating 70,175 bp of core chromosome 11 plus 15–87 kb of paralogous sub-telomeric region replacing the chromosomal region on chromosome 13 that contained *pfhrp3*. Importantly, NAHR-mediated translocations resulting in deletion have repeatedly occurred based on evidence of multiple breakpoints and chromosome 11 haplotypes with identical copies in parasites. These findings, combined with identical copies of the shared chromosome 11 segment, suggest that these parasites represent multiple instances of intrastrain (self) NAHR-mediated translocation followed by clonal propagation of 13$^-$11$^{++}$ progeny. Within haplotype groups with identical chromosome 11 duplicated regions, there appear to be several clonal groups, which suggests that different events lead to the same duplication of chromosome 11, and each continued to have clonal propagation (*Figure 1—figure supplement 5*). While Hinterberg et al. proposed that a general mechanism of non-homologous recombination of the subtelomeric regions may be responsible for translocating the already existing deletion of *pfhrp3* (*Hinterberg et al., 1994*), our analysis would suggest ribosomal duplication-mediated NAHR is the likely cause of the *pfhrp3* deletion itself. The high frequency of the meiotic translocation in the laboratory cross further supports the hypothesis that these NAHR-mediated translocations are occurring at a high frequency in meiosis in natural populations. Consequently, this suggests that progeny must be strongly selected against in natural populations apart from where specific conditions exist, allowing for *pfhrp3* deletion to emerge and expand (e.g. South America and the Horn of Africa).

Positive selection due to drug resistance may underlie pattern 13$^-$5$^{++}$ translocation that duplicates *pfmdr1* onto chromosome 13. In South East Asia (SEA), the only place containing pattern 13$^-$5$^{++}$, tandem duplications of *pfmdr1* exist that provide mefloquine resistance, and mefloquine has been used extensively as an artemisinin partner drug, unlike Africa (*Veiga et al., 2016*). Discordant reads, local assembly, and TARE1 identification support NAHR-mediated translocation of *pfmdr1*, followed by telomeric healing to create a functional chromosome. All strains showed the same exact NAHR breakpoint and TARE1 localization consistent with a single origin event, giving rise to all 13$^-$5$^{++}$ parasites. Interestingly, *pfmdr1* duplications have been shown to be unstable, with both increases and decreases in copy numbers frequently occurring (*Samarakoon et al., 2011*). During de-amplification, a free fragment of DNA containing a *pfmdr1* copy may have been the substrate that integrated into chromosome 13 by NAHR, followed by telomerase healing to stabilize the 13–5 hybrid chromosome, analogous to *var* gene recombination events where double-stranded DNA is displaced, becoming highly recombinogenic (*Zhang et al., 2019*). The spread of 13$^-$5$^{++}$ containing parasites could be due to the benefit of the extra *pfmdr1* copy on chromosome 13, the loss of *pfhrp3*, or both. Its expansion in SEA would be consistent with selection due to copy-number-associated mefloquine resistance, given mefloquine's extensive use as an individual and artemisinin partner drug in the region. Furthermore, given all isolates with evidence of this duplication either had only wild-type *pfmdr1* or were mixed, the 13–5 chromosome copy most likely had wild-type *pfmdr1*. In 20 out of 25 *pfmdr1* duplication cases with a mixed genotype of *pfmdr1* (*Figure 1—figure supplement 5*), the core genome *pfmdr1* had the Y184F mutation with no other mutations detected within the *pfmdr1* gene. Isolates containing only Y184F in *pfmdr1* were shown to be outgrown by wild-type *pfmdr1* (*Duvalsaint et al., 2021*), which would mean having the wild-type *pfmdr1* duplication on chromosome 13 might confer a

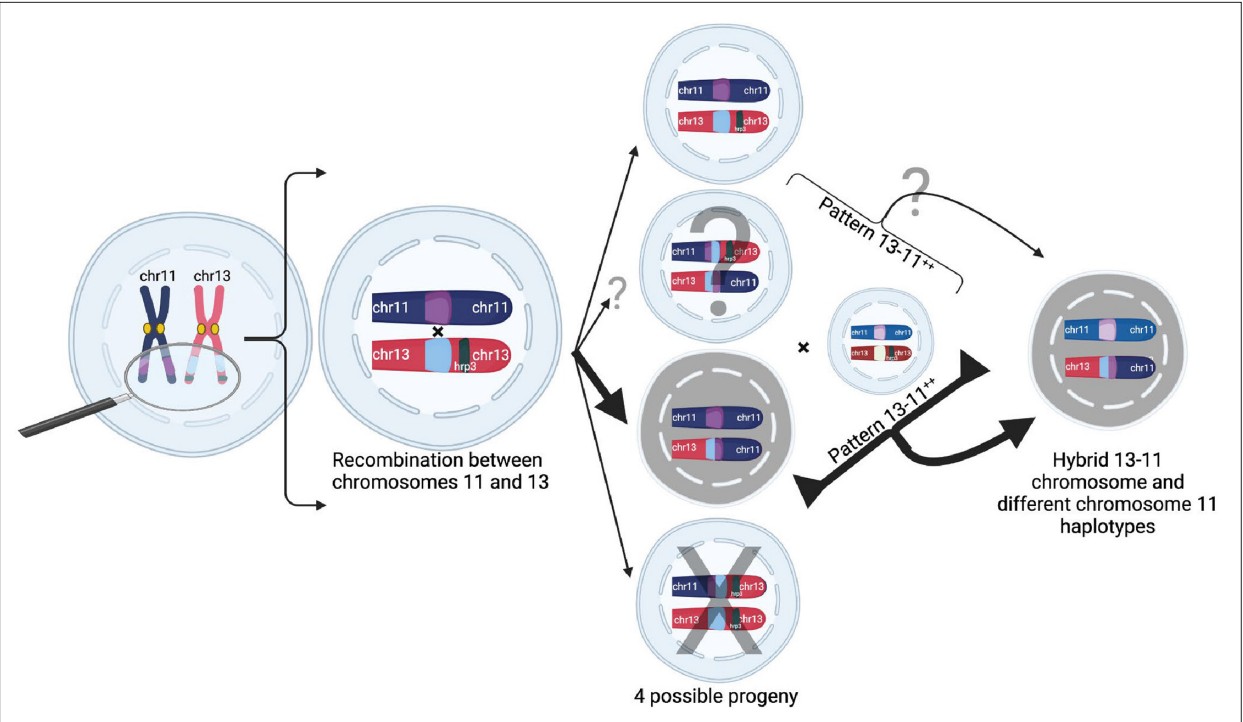

**Figure 6.** Proposed model of duplication-mediated non-allelic homologous recombination during intrastrain meiotic recombination yielding $13^-11^{++}$ parasites. Homology misalignment and non-allelic homologous recombination (NAHR) between chromosomes 11 and 13 first occur in an oocyst formed from identical parasite gametes (intrastrain), which can then segregate, resulting in potential progeny (normal and three translocated progeny). Bold lines show the most direct path to a $13^-11^{++}$ parasite containing a 13–11 hybrid chromosome lacking *pfhrp3* and two identical copies of duplicated chromosome 11 segments seen predominantly. Subsequent recombination with an unrelated strain yields parasites with differing chromosome 11 duplication haplotypes but this can occur with subsequent interstrain meioses. Additionally, there is potential for balanced products, occurring with subsequent recombination events leading to *pfhrp3* loss and either identical haplotypes (intrastrain) or different haplotypes (unrelated strain). Figure created using Biorender.

stable (non-tandem) 'heterozygous' survival advantage beyond just increased copy number-mediated resistance to mefloquine.

To confirm the NAHR event between 11 and 13 leading to loss of *pfhrp3* observed in our analysis of short-read data, we long-read sequenced *pfhrp3* deleted lab isolates, HB3 and SD01, to generate reads spanning the 15 kb duplicated region, showing support for a normal chromosome 11 and a hybrid 13–11 in both isolates. These findings supported an NAHR event between the two 15 kb duplicated regions, causing this interchromosomal exchange and leading to progeny with a hybrid 13–11 chromosome lacking *pfhrp3* and its surrounding genes from the 15 kb duplicated region and onwards (*Figure 4* and *Figure 5*). This was consistent with the genomic coverage pattern we observed in publicly available data from 117 *pfhrp3*-deleted field samples (*Figure 1*). Such translocation patterns have been described and confirmed by long-read sequencing but have generally involved multigene families such as *var* genes within the subtelomeres (*Calhoun et al., 2017*; *Zhang et al., 2019*). The event described here represented a much larger section of loss/duplication of 70 kb of the core genome in addition to the subtelomere.

We propose a mechanistic model in which homology misalignment and recombination between chromosomes 11 and 13 initially occurs in an oocyst from identical parasites predominately in low transmission settings, resulting in four potential progeny including one with normal chromosomes 11 and 13 and three with translocations (*Figure 6*). This could account for the identical haplotypes observed in the two copies of the chromosome 11 segment. Based on the identical haplotypes observed in most parasites, the most direct and likely mechanism involves progeny with two copies of chromosome 11 recombining with an unrelated strain to yield unrelated chromosome 11 haplotypes. This duplication-mediated NAHR event occurs frequently during meiosis and can explain the frequent rearrangements seen between chromosomes 11 and 13 in the previous experimental cross of HB3

× DD2 (*Hinterberg et al., 1994*). Meiotic misalignment and subsequent NAHR is a common cause of high-frequency chromosomal rearrangements, including in human disease (e.g. in humans, 22q11 deletion syndrome due to misalignment of duplicated blocks on chromosome 22 occurs in 1 in 4000 births)(*Vergés et al., 2014*). This high frequency could explain why *pfhrp3*-deleted isolates are more common in many populations relative to *pfhrp2*(*Feleke et al., 2021*; *Góes et al., 2020*; *Vera-Arias et al., 2022*), which likely requires infrequent random breaks along with rescue by telomere healing. In the future, more extensive sequencing of RDT-negative *P. falciparum* parasites is needed to confirm that no other deletion mechanisms are responsible for *pfhrp2* loss.

The lack of hybrid chromosome 13–11 (pattern 13$^-$11$^{++}$) worldwide suggests such events are normally quickly removed from the population due to fitness costs, an idea supported by recent in vitro competition studies in culture showing decreased fitness of *pfhrp2/3*-deleted parasites (*Nair et al., 2022*). This decreased fitness of parasites with *pfhrp2/3* deletions also argues against a mitotic origin, as deletions arising after meiosis would have to compete against more numerous and more fit intact parasites. Additionally, *pfhrp3* deletions arising in culture have not been observed. The fact that abundant *pfhrp3* deletions have only been observed in low-transmission areas where within-infection competition is rare is consistent with this hypothesis of within-infection competition suppressing emergence. In the setting of RDT use in a low-transmission environment, a pfhrp2 deletion occurring in the context of an existing *pfhrp3* deletion may be more strongly selected for compared to a *pfhrp2* deletion occurring alone still detectable by RDTs. This is supported by evidence that *pfhrp3* deletion appears to predate *pfhrp2* deletions in the Horn of Africa (*Feleke et al., 2021*).

The biological effects of *pfhrp2* and *pfhrp3* loss and potential selective forces are complicated due to other genes lost and gained and the extent of the rearrangements. Increased copies of genes on chromosome 11 could be beneficial, as *pf332* on the chromosome 11 duplicated segment was found to be essential for binding the Maurer cleft to the erythrocyte skeleton and is highly expressed in patients with cerebral malaria (*Bertin et al., 2016*). Conversely, the lack of this protein is likely detrimental to survival and may be why the reciprocal hybrid 11–13 (pattern 13$^{++}$11$^-$) was not observed in field isolates. Only lab isolate FCR3 had any indication from coverage data that it had a duplicated chromosome 13 and a deleted chromosome 11. Furthermore, when looking at the field samples with chromosome 11 deletions with telomere healing, breakages were almost exclusively right after *pf332* (*Figure 5—figure supplement 6*), which suggests this protein is vital. Given that the majority of the publicly available field samples were collected from studies using RDT-positive samples and that RDT would have likely detected the increased PfHRP3 encoded by duplicated *pfhrp3*, sampling should not be biased against detecting parasites with this reciprocal hybrid 11–13. Thus, the lack of 11–13 rearrangement in field isolates suggests that the selective disadvantage of the lost and gained genes was strong enough to prevent its emergence in the natural parasite population.

While further studies are needed to determine the reasons for these geographical patterns of *pfhrp3* deletions, our results provide an improved understanding of the mechanism of structural variation underlying *pfhrp3* deletion. They also suggest general constraints against emergence in high-transmission regions due to within-host competition and that there are likely further specific requirements for emergence in low-transmission settings. If selective constraints of *pfhrp2* and *pfhrp3* deletions are similar, the high frequency of the NAHR-mediated loss and the additional drug pressure from duplication of *pfmdr1* may explain why *pfhrp3* loss precedes *pfhrp2* loss despite RDT pressure presumably exerting stronger survival advantage with loss of *pfhrp2* versus *pfhrp3*. However, given we still have a limited understanding of their biological roles, there may be situations where selective forces may favor loss of *pfhrp2* relative to *pfhrp3*. Limitations of this study include the use of publicly available sequencing data that were collected often based on positive rapid diagnostic tests, which limits our interpretation of the occurrence and relative frequency of these deletions. This could introduce regional biases due to different diagnostic methods as well as limit the full range of deletion mechanisms, particularly *pfhrp2*. Our findings are clinically important because the continued loss of these genes without timely intervention may result in a rapid decrease in the sensitivity of HRP2-based RDTs. To the best of our knowledge, no prior studies have performed long-read sequencing to definitively span and assemble the entire segmental duplication involved in the deletions. Future studies focused on these deletions, including representative sampling, are needed to determine the prevalence, interactions, and impacts of *pfhrp2* and *pfhrp3* deletions and the selective pressures and complex biology underlying them.

## Materials and methods
### Genomic locations used
Conserved non-paralogous genomic regions surrounding *pfhrp2* and *pfhrp3* were determined to study the genomic deletions encompassing these genes. This was accomplished by first marking the 3D7 genome with the program tandem repeat finder (*Benson, 1999*), then taking 200 bp windows, stepping every 100 bp between these tandem repeats, and using LASTZ (*Harris, 2007*)(version 1.04.22) to align these regions against the reference genome 3D7 (version 3, 2015-06-18) and 10 currently available chromosomal-level PacBio assembled genomes (*Otto et al., 2018a*) that lacked *pfhrp2* and *pfhrp3 deletions*. Regions that aligned at >70% identity in each genome only once were kept, and overlapping regions were then merged. Regions within the duplicated region on chromosome 11 and chromosome 13 were kept if they aligned to either chromosome 11 or 13 but not to other chromosomes. This region selection method decreases artifacts from multi-mapping and discordant reads. It allows analysis of only regions present in most strains, avoiding regions that may be absent from telomeric rearrangements. In this way, we have selected regions that can be reliably used to look for signals in coverage to detect copy number variation by minimizing the amount of contribution from artifacts from erroneous read mapping.

### Whole genome sequencing analysis, assembly, read-depth analysis, and variant calling
Data was collected from all publicly available Illumina whole-genome sequencing (WGS) data from global *P. falciparum* isolates as of January 2023, comprising 19,313 field samples of which 33 were lab isolates (*Supplementary file 1*). The majority of the data came from the Pf7 project (*Abdel Hamid et al., 2023*) but also included several smaller sequencing projects (*Cerqueira et al., 2017*; *Dara et al., 2017*; *Feleke et al., 2021*; *LaVerriere et al., 2022*; *Mathieu et al., 2020*; *Melnikov et al., 2011*; *Parobek et al., 2017*; *Pelleau et al., 2015*; *Tvedte et al., 2021*; *Villena et al., 2021*). Raw fastqs were downloaded using their short read archive (SRA) accession numbers. Using default settings, raw reads were trimmed with AdapterRemoval (*Schubert et al., 2016*). Processed reads were mapped to a genome file that was created by combining the human genome (GRCh38), *mycoplasma* genomes [Genbank accession AP014657.1 (*Hata, 2015*) and LR214940.1], and 3D7 *Plasmodium falciparum* (version 2015-06-18) genome with bwa-mem (*Li, 2013*) using settings of '-M -k 25' to optimize for contamination removal (*Robinson et al., 2017*). Reads that mapped to the human genome or *mycoplasma* genomes were removed to remove the human contamination from the field WGS data and to remove the *mycoplasma* contamination in the lab isolates. Filtered reads were then remapped to the 3D7 *Plasmodium falciparum* (version 2015-06-18) genome using the default settings of bwa-mem (*Li, 2013*).

Read depth for the above regions was then determined by taking the reads mapped to the region of interest and adjusting per-base read depth for read pairs (if both mates covered a specific base, its read coverage contribution was only 1 rather than 2). Read depth was then normalized by dividing by the median coverage of the whole genome. Read depth for the regions shared between chromosomes 11 and 13 was determined by taking reads mapped to the corresponding regions on 11 or 13 and then dividing by 2 to get the coverage for those regions. Samples were eliminated for downstream analysis if they had less than 5 x median genomic coverage and sWGA samples with significant variability on visible inspection. For most visualizations, the coverage is rounded to the nearest 1, and the max coverage is capped at 2 x coverage for better visualization. To ensure analysis of only highly confident deleted strains, normalized read depth between 0.1–1 was rounded to 1, and less than 0.1 was rounded to 0. Samples were considered for potential genomic deletion if they had zero coverage after rounding from chromosome 8 1,375,557–1,387,982 for *pfhrp2*, chromosome 13 from 2,841,776–2,844,785 for *pfhrp3*, and from chromosome 11 1,991,347–2,003,328. These numbers were chosen after visual inspection of samples with any zero coverage within the genomic region of *pfhrp2/3*. Looking for potential deletions by utilizing zero coverage limits detection of deletion only to monoclonal samples or mix-infection where all strains have deletions.

Variant calling was performed by GATK version 4.3.0 (*Poplin et al., 2018*), with default settings only on the non-paralogous regions, as determined above. Variants were then filtered to only biallelic single nucleotide polymorphisms (SNPs). The complexity of infection (COI) was determined using the REAL McCOIL (*Chang et al., 2017*) on the 1000 highest expected heterozygosity SNPs. Identity by

descent was calculated using hmmIBD (*Schaffner et al., 2018*) to get inter-strain relatedness between samples.

Local haplotype reconstruction was also performed on each region described above on all the WGS samples using PathWeaver (https://github.com/nickjhathaway/PathWeaver copy archived at *Hathaway, 2024* manuscript in preparation). PathWeaver takes the reads mapped to a region of interest and then performs a de Bruijn graph construction while keeping read information to inform graph traversal to prevent chimera assemblies in polyclonal samples. The haplotypes from Path-Weaver were analyzed by constructing a graph for each genomic region to determine the conserved subregions that all haplotypes traversed using custom C++ scripts. Smaller variable regions were then determined to be those between these conserved regions. These variable subregions were used for genotyping by running PathWeaver again on these new regions. The regions analyzed were chromosome 05 (including *pfmdr1*): 929,384–988,747,, chromosome 08: 1,290,239–1,387,982, chromosome 11: 1,897,151–2,003,328, chromosome 13: 2,769,916–2,844,785 (*Figure 5—figure supplement 7*, *Figure 5—figure supplement 8*). To validate the PathWeaver results, we also analyzed these regions using the biallelic SNPs called by GATK as described above within the regions of interest (*Supplementary file 4*). The biallelic SNPs analysis produced similar results (*Figure 5—figure supplement 9*, *Figure 5—figure supplement 10*), and we opted to use the PathWeaver results as they give finer resolution by analyzing at a haplotype level rather than an SNP level. All coordinates in the manuscript are zero-based positioning relative to *P. falciparum* 3D7 genome version 3 (version = 2015-06-18) (*Figure 5—figure supplement 7*, *Figure 5—figure supplement 8*).

## Tandem repeat associated element 1 (TARE1) analysis and telomere healing determination

The 7 bp pattern of TT[CT]AGGG (*Vernick and McCutchan, 1988*) was used to determine the presence of TARE1, of which the presence of this pattern was required to occur at least twice in tandem. To search for the presence of TARE1 within the short read Illumina WGS datasets, reads from the entire regions of interest were pulled down across chromosomes 5, 8, 11, and 13, and the above TARE1 pattern was searched for in each read. Regions with TARE1 detected in their reads were assembled to ensure the TARE1 sequence was contiguous with the genomic region from which the reads were pulled. The regions with TARE1 contiguous with genomic regions were then compared to the coverage pattern within the area. A parasite was marked as having evidence of telomere healing if TARE1 was detected in the regions where coverage then dropped to 0 or down to the genomic coverage of the rest of the genome in the case of the chromosome 5 duplication.

## Chromosome 5 pfmdr1 duplication breakpoint determination

The recombination breakpoints on chromosome 5 were determined by looking for discordant read pairs (mates mapping to different chromosomal positions around areas of interest), assembling the discordant pairs, and mapping back to the assembled contig. Breakpoints were then determined by looking at the coordinates where a contig switches from one chromosomal region to the next. This was done by utilizing custom C++ scripts. The genomic regions where coverage had a sudden fold increase to 2 x or 3 x coverage were scanned in 500 bp windows, and discordant mates, as determined by their alignment by bwa, were gathered in each window. If a window had several discordant mates that mapped to within 500 bp of each other, these regions would be linked. Raw reads were then pulled from the original and discordant regions and assembled using PathWeaver. The assembled contigs were then aligned using LASTZ to determine where a contig switched from one chromosomal region (upstream for the tandem duplication or to a different chromosome like the chromosome 13 translocation) to determine the exact genomic locations of the breakpoints.

## pfhrp3 pattern determination

A sample was considered pattern 13⁻11⁺⁺ if it had zero coverage on chromosome 13 from the chromosome 11/13 duplicated region with no detectable TARE1 sequence at the location where coverage dropped to zero and if it had detectable coverage on chromosome 11 from the duplicated region onwards. Pattern 13⁻5⁺⁺ was determined if a sample had discordant mates mapping from the region on chromosome 13, where that sample dropped to zero, to chromosome 5. Pattern 13⁻TARE1 was

determined if a sample had a TARE1 sequence detected as contiguous with chromosome 13 sequence at the location on chromosome 13, where that sample dropped to zero.

## Homologous genomic structure

To investigate the genomic landscape of recent segmental duplications across the genome and around *pfhrp2 and pfhrp3*, an all-by-all comparison of the 3D7 reference genome was performed by using the program nucmer and mummer (*Kurtz et al., 2004*). These programs align genomes by using unique kmers found within the genomes. To analyze the 3D7 reference genome, we separated the chromosomes into individual fasta files and used nucmer and mummer in an all-by-all comparison. Nucmer was run with the following commands: 'nucmer -mum -b 100 l 31 [CHROM_1].fasta [CHROM_2].fasta' and the following commands when comparing a chromosome against itself 'nucmer –reverse -mum -b 100 l 31 [CHROM_1].fasta [CHROM_1].fasta' to compare against only it's reverse complement. Mummer was run with the following commands: 'mummer -mum -b -c -F -l 31 [CHROM_1].fasta [CHROM_2].fasta' and when comparing a chromosome against itself 'mummer -mum -r -c -F -l 31 [CHROM_1].fasta [CHROM_1].fasta.' The nucmer results can be found in *Supplementary file 5*.

## Comparisons within *Laverania*

To investigate the origins of this region shared between chromosomes 11 and 13, the six closest relatives of *Plasmodium falciparum* within the *Laverania* subgenus with available assembled genomes were examined (*Otto et al., 2018b*). The genomes of all *Laverania* have recently been sequenced and assembled using PacBio and Illumina data (*Otto et al., 2018b*). The assemblies were analyzed using their annotations and LASTZ (*Harris, 2007*) with 80% identity and 90% coverage of the genes in the surrounding regions on chromosomes 5, 8, 11, and 13.

## Long-read sequences

All PacBio reads for strains with known or suspected *pfhrp3* deletions were obtained by SRA accession numbers from the National Center for Biotechnology Information (NCBI): HB3/Honduras (ERS712858) and SD01/Sudan (ERS746009)(*Otto et al., 2018a*). To supplement these reads and to improve upon previous assemblies that were unable to fully assemble chromosomes 11 and 13, we further sequenced these strains using Oxford Nanopore Technologies' MinION device (*Branton et al., 2008*; *Jain et al., 2016*; *Kasianowicz et al., 1996*). The *P. falciparum* lab isolates HB3/Honduras (MRA-155) was obtained from the National Institute of Allergy and Infectious Diseases' BEI Resources Repository. The field strain SD01/Sudan was obtained from the Department of Cellular and Applied Infection Biology at Rheinisch-Westfälische Technische Hochschule (RWTH) Aachen University in Germany. Nanopore base-calling was done with Guppy version 5.0.7. Genome assemblies were performed with Canu (*Koren et al., 2017*), an assembly pipeline for high-noise single-molecule sequencing, and Flye (*Kolmogorov et al., 2019*) using default settings. To assemble the low coverage and highly similar chromosome 11 and 13 segments of SD01, two assemblies were performed with Flye using chromosome 13-specific and 11-specific reads to get contigs representing the chromosome 11 and 13 segments (*Figure 4—figure supplement 2*) and final contigs were polished using the Illumina SD01 reads using the program pilon (*Walker et al., 2014*). HB3 was assembled using the Canu assembler with default settings. Note that SD01 had a more disjointed assembly, likely due to coming from the last remaining cryopreserved vial, which had low parasitemia, was nonviable, and had a subsequent lower amount of input DNA. The PacBio/Nanopore reads were mapped to the 3D7 reference genome with hybrid chromosomes 11–13 and 13–11 using Minimap2 with default settings, a sequence alignment program (*Li, 2018*). Mappings to the duplicated region between chromosomes 11 and 13 were visualized using custom R scripts (*Gu, 2022*; *R Development Core Team, 2022*; *Wickham et al., 2022*). Reads were considered to be spanning the duplicated region if they extended at least 50 bp upstream and downstream from the duplicated region and if they mapped only to that one region. Non-spanning reads mapped to normal chromosomes 11 and 13 and the hybrid chromosomes as these genomic segments are identical.

## Acknowledgements

We thank Drs. Ngwa Julius Che, Matthias Frank, and Gabriele Pradel from Rheinisch-Westfälische Technische Hochschule (RWTH) Aachen University for generously providing a residual SD01 sample.

The following reagent was obtained through BEI Resources, NIAID, NIH: *Plasmodium falciparum*, Strain HB3, MRA-155, contributed by Thomas E Wellems. We thank the National Institutes of Allergy and Infectious Diseases (NIAID) for their support via the grants R01AI132547 (JJJ JBP and JAB) and K24AI134990 (JJJ)

## Additional information

### Funding

| Funder | Grant reference number | Author |
|---|---|---|
| National Institute of Allergy and Infectious Diseases | R01AI132547 | Jonathan J Juliano<br>Jonathan B Parr<br>Jeffrey A Bailey |
| National Institute of Allergy and Infectious Diseases | K24AI134990 | Jonathan J Juliano |

The funders had no role in study design, data collection and interpretation, or the decision to submit the work for publication.

### Author contributions

Nicholas J Hathaway, Isaac E Kim, Conceptualization, Data curation, Formal analysis, Investigation, Visualization, Methodology, Writing – original draft, Writing – review and editing; Neeva Wernsman-Young, Rebecca Crudale, Emily Y Liang, Christian P Nixon, David Giesbrecht, Investigation, Methodology, Writing – original draft, Writing – review and editing; Sin Ting Hui, Investigation, Visualization, Methodology, Writing – original draft, Writing – review and editing; Jonathan J Juliano, Jonathan B Parr, Supervision, Funding acquisition, Validation, Investigation, Writing – original draft, Writing – review and editing; Jeffrey A Bailey, Supervision, Funding acquisition, Investigation, Writing – original draft, Writing – review and editing

### Author ORCIDs

Nicholas J Hathaway (ID) http://orcid.org/0000-0001-9639-2894
Isaac E Kim (ID) https://orcid.org/0000-0002-9737-8100
Jeffrey A Bailey (ID) https://orcid.org/0000-0002-6899-8204

Reviewer #1 (Public review): https://doi.org/10.7554/eLife.93534.3.sa1
Reviewer #2 (Public review): https://doi.org/10.7554/eLife.93534.3.sa2
Reviewer #3 (Public review): https://doi.org/10.7554/eLife.93534.3.sa3
Author response https://doi.org/10.7554/eLife.93534.3.sa4

## Additional files

### Supplementary files

• Supplementary file 1. Metadata on *P. falciparum* samples. This table includes information on the publicly available, whole-genome-sequenced samples used for primary analysis including the identification number, sample name, country, collection year and date, region, and pfhrp2/3 deletion pattern.

• Supplementary file 2. Core genome windows. This table contains information on the non-paralogous windows used for analyzing the regions of interest on chromosomes 5, 7, 8, 11, and 13. *Plasmodium falciparum* samples. Information includes the window's chromosome name in 3D7, starting base position, ending base position, length, which strand the window is in respect to (positive or negative DNA strand), and description of the window region's functionality.

• Supplementary file 3. Variations in core genome windows. This table contains information on the windows contianing variations in the non-paralogous windows used for analyzing the regions of interest on chromosomes 5, 7, 8, 11, and 13. These variable regions can be found completely within the windows of *Supplementary file 1*. Metadata on *P. falciparum* samples.

• Supplementary file 4. Biallelic SNPs. This table contains information on the biallelic SNPS from within the non-paralogous windows used for analyzing the regions of interest on chromosomes 5, 7, 8, 11, and 13. These variable regions can be found completely within the windows of *Supplementary file 2*. Core genome windows and *Supplementary file 3*. Variations in core genome windows.

• Supplementary file 5. Nucmer Results. This table contains the results of the nucmer analysis looking for inter-chromosomal exact sequence sharing. Majority of the shared sequence is within the sub-telomere regions with the largest 'core' genomic region of a shared sequence is the 11–13 rRNA loci.

• Supplementary file 6. Study Accession IDs.

• MDAR checklist

## Data availability

Nanopore data is available from the SRA (Project PRJNA1131468 (HB3), Project PRJNA1131459 (SD01)). The datasets analyzed during the current study and the code used to analyze them are available at https://doi.org/10.5281/zenodo.12167687. The study accession IDs of all previously published datasets used in this study are included in *Supplementary file 6*.

The following datasets were generated:

| Author(s) | Year | Dataset title | Dataset URL | Database and Identifier |
|---|---|---|---|---|
| Hathaway N | 2024 | Interchromosomal segmental duplication drives translocation and loss of *P. falciparum* histidine-rich protein 3 support documentation | https://doi.org/10.5281/zenodo.12167686 | Zenodo, 10.5281/zenodo.12167686 |
| Brown University | 2024 | SRX25199648: Nanopore Sequencing of *Plasmodium falciparum* isolate HB3 | https://www.ncbi.nlm.nih.gov/sra/SRX25199648 | NCBI Sequence Read Archive, SRX25199648 |
| Brown University | 2024 | SRX25198112: *Plasmodium falciparum* isolate SD01 | https://www.ncbi.nlm.nih.gov/sra/SRX25198112 | NCBI Sequence Read Archive, SRX25198112 |

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
