## [Editor Report · eLife assessment]

This work provides **important** insight into the mechanisms of hrp2 and particularly hrp3 deletion generation. The generation of additional long-read data alongside a new analysis of 19,000 public short-read sequenced genomes makes this the most detailed investigation currently available on this topic, which has high public health importance and also basic biological interest. The revised version of the manuscript provides **convincing** evidence for the proposed mechanisms.

---

## [Referee Report · Reviewer #1 (Public review)]

Summary:

Deletion of the hrp2 and hrp3 loci in *P. falciparum* poses an immediate public health threat. This manuscript provides a more complete understanding of the dynamic nature with which these deletions are generated. By delving into the likely mechanisms behind their generation, the authors also provide interesting insight into general Plasmodium biology that can inform our broader understanding of the parasite's genomic evolution.

Strengths:

The sub-telomeric regions of *P. falciparum* (where hrp2 and hrp3 are located) are notoriously difficult to study with short-read sequence data. The authors take an appropriate, targeted approach toward studying the loci of interest, which includes read-depth analysis and local haplotype reconstruction. They additionally use both long-read and short-read data to validate their major findings. There is an extensive set of supplementary plots, which helps clarify several aspects of the data.

Weaknesses:

The revised version of this manuscript has helpfully expanded the details regarding methodology, however, publication of the tool PathWeaver (which is used for local haplotype reconstruction) remains in preparation.

---

## [Referee Report · Reviewer #2 (Public review)]

This work investigates the mechanisms, patterns and geographical distribution of pfhrp2 and pfhrp3 deletions in Plasmodium falciparum. Rapid diagnostic tests (RDTs) detect *P. falciparum* histidine-rich protein 2 (PfHRP2) and its paralog PfHRP3 located in subtelomeric regions. However, laboratory and field isolates with deletions of pfhrp2 and pfhrp3 that can escape diagnosis by RDTs are spreading in some regions of Africa. They find that pfhrp2 deletions are less common and likely occurs through chromosomal breakage with subsequent telomeric healing. Pfhrp3 deletions are more common and show three distinct patterns: loss of chromosome 13 from pfhrp3 to the telomere with evidence of telomere healing at breakpoint (Asia; Pattern 13-); duplication of a chromosome 5 segment containing pfhrp1 on chromosome 13 through non-allelic homologous recombination (NAHR) (Asia; Pattern 13-5++); and the most common pattern, duplication of a chromosome 11 segment on chromosome 13 through NAHR (Americas/Africa; Pattern 13-11++). The loss of these genes impact the sensitivity od RDTs, and knowing these patterns and geographic distribution makes it possible to make better decisions for malaria control.

Comments on latest version:

The authors answered all my questions.

---

## [Referee Report · Reviewer #3 (Public review)]

The study provides a detailed analysis of the chromosomal rearrangements related to the deletions of histidine-rich protein 2 (pfhrp2) and pfhrp3 genes in *P. falciparum* that have clinical significance since malaria rapid diagnostic tests detect these parasite proteins. A large number of publicly available short sequence reads for whole-genome of the parasite were analyzed and data on coverage and on discordant mapping allowed to identify deletions, duplications and chromosomal rearrangements related to pfhrp3 deletions. Long-read sequences showed support for the presence of a normal chromosome 11 and a hybrid 13-11 chromosome lacking pfhrp3 in some of the pfhrp3-deleted parasites. The findings support that these translocations have repeatedly occurred in natural populations. The authors discuss the implications of these findings and how they support or not previous hypothesis on the emergence of these deletions and the possible selective pressures involved.

The genomic regions where these genes are located are challenging to study since they are highly repetitive and paralogous and the use of long read sequencing allowed to span the duplicated regions, giving support to the identification of the hybrid 13-11 chromosome.

All publicly available whole-genome sequences of the malaria parasite from around the world were analysed which allowed an overview of the worldwide variability, even though this analysis is biased by the availability of sequences, as the authors recognize.

Despite the reduced sample size, the detailed analysis of haplotypes and identification of location of breakpoints gives support to a single origin event for the 13-5++ parasites.

The analysis of haplotype variation across the duplicated chromosome-11 segment identified breakpoints at varied locations that support multiple translocation events in natural populations. The authors suggest these translocations may be occurring at high frequency in meiosis in natural populations but strongly selected against in most circumstances, which remains to be tested.

In this new version, the authors have addressed the points raised previously and adequately discuss the limitations of the study.

---

## [Author Response]

The following is the authors’ response to the original reviews.

**Reviewer #1 (Public Review):**
Summary:Deletion of the hrp2 and hrp3 loci in *P. falciparum* poses an immediate public health threat. This manuscript provides a more complete understanding of the dynamic nature with which these deletions are generated. By delving into the likely mechanisms behind their generation, the authors also provide interesting insight into general Plasmodium biology that can inform our broader understanding of the parasite's genomic evolution.Strengths:The sub-telomeric regions of *P. falciparum* (where hrp2 and hrp3 are located) are notoriously difficult to study with short-read sequence data. The authors take an appropriate, targeted approach toward studying the loci of interest, which includes read-depth analysis and local haplotype reconstruction. They additionally use both long-read and short-read data to validate their major findings. There is an extensive set of supplementary plots, which helps clarify several aspects of the data.Weaknesses:In this first version, there are a few factors that hinder a full assessment of the robustness and replicability of the results.
**Reviewer #1 (Recommendations For The Authors):**
Reviewer comment: First, a number of the analyses lack basic details in the methods; for instance, one must visit the authors' personal website to find some of the tools used.

We have extensively updated the methods to clarify which tools were used and how they were run. All code and results for the analyses have been deposited in Zenodo at https://doi.org/10.5281/zenodo.12167687.

Reviewer comment: Second, there are several tricky methodological points that are not fully documented. Read depths are treated (and plotted) discretely as 0/1/2 without any discussion of how thresholds were used and determined.

We have added to the methods section the full details on how read depth was handled, including rounding to the closest 1 normalized coverage for visualizations. To ensure analysis of only highly confident deleted strains, normalized coverage of 0.1 or more was round to 1 instead of 0. Samples were considered for potential genomic deletion if they had zero coverage after rounding from chromosome 8 1,375,557 to 1,387,982 for *pfhrp2*, chromosome 13 from 2,841,776 to 2,844,785 for *pfhrp3*, and from chromosome 11 1,991,347 to 2,003,328. These numbers were chosen after visual inspection of samples with any zero coverage within the genomic region of *pfhrp2/3*.

Reviewer comment: For read mapping to standard vs hybrid chromosomes, there is no documentation on how assignments were made if partially ambiguous or how final sample calls were determined when some reads were discordant. There is no mention of how missing data were handled. Without this, it is difficult to know when conclusions were based on analyses that were more quantitative (for instance, using pre-determined read thresholds) or more subjective (with patterns being extracted visually).

We have updated several parts of the methods section to explicitly state what thresholds and analysis pipelines to use, making our documentation clearer. For mapping to the hybrid vs standard chromosomes for the long reads, spanning reads across the duplicated region were required to extend 50bp upstream and downstream of the region. These regions are significantly different between chromosomes 11 and 13, so requiring spanning reads to map to these regions prevented multi-mapping reads. Reads that started within the duplicated region were allowed to map to both the hybrid and standard chromosomes for visualization in Figure 4. Importantly, for both HB3 and SD01, no reads spanned from the duplicated region into chromosome 13, showing a complete lack of reads that contained the portion of chromosome 13 that came after the duplicated region. None of the other isolates had any spanning reads across the hybrid chromosomes. Details on deletion calls were based on initial visualization of *pfhrp2/3* and then on read thresholds (see above response for details).

Reviewer comment: Third, while a new method is employed for local haplotype reconstruction (PathWeaver), the manuscript does not include details on this approach or benchmarking data with which to evaluate its performance and understand any potential artifacts.

We have added an analysis based on biallelic SNPs to compare to the PathWeaver results, which produced similar results to help validate the PathWeaver results. PathWeaver manuscript is in preparation.

**Reviewer #2 (Public Review):**
This work investigates the mechanisms, patterns, and geographical distribution of pfhrp2 and pfhrp3 deletions in Plasmodium falciparum. Rapid diagnostic tests (RDTs) detect *P. falciparum* histidine-rich protein 2 (PfHRP2) and its paralog PfHRP3 located in subtelomeric regions. However, laboratory and field isolates with deletions of pfhrp2 and pfhrp3 that can escape diagnosis by RDTs are spreading in some regions of Africa. They find that pfhrp2 deletions are less common and likely occur through chromosomal breakage with subsequent telomeric healing. Pfhrp3 deletions are more common and show three distinct patterns: loss of chromosome 13 from pfhrp3 to the telomere with evidence of telomere healing at breakpoint (Asia; Pattern 13-); duplication of a chromosome 5 segment containing pfhrp1 on chromosome 13 through non-allelic homologous recombination (NAHR) (Asia; Pattern 13-5++); and the most common pattern, duplication of a chromosome 11 segment on chromosome 13 through NAHR (Americas/Africa; Pattern 13-11++). The loss of these genes impacts the sensitivity of RDTs, and knowing these patterns and geographic distribution makes it possible to make better decisions for malaria control.
**Reviewer #3 (Public Review):**
Summary:The study provides a detailed analysis of the chromosomal rearrangements related to the deletions of histidine-rich protein 2 (pfhrp2) and pfhrp3 genes in *P. falciparum* that have clinical significance since malaria rapid diagnostic tests detect these parasite proteins. A large number of publicly available short sequence reads for the whole genome of the parasite were analyzed, and data on coverage and discordant mapping allowed the authors to identify deletions, duplications, and chromosomal rearrangements related to pfhrp3 deletions. Long-read sequences showed support for the presence of a normal chromosome 11 and a hybrid 13-11 chromosome lacking pfhrp3 in some of the pfhrp3-deleted parasites. The findings support that these translocations have repeatedly occurred in natural populations. The authors discuss the implications of these findings and how they do or do not support previous hypotheses on the emergence of these deletions and the possible selective pressures involved.Strengths:The genomic regions where these genes are located are challenging to study since they are highly repetitive and paralogous and the use of long-read sequencing allowed to span the duplicated regions, giving support to the identification of the hybrid 13-11 chromosome.All publicly available whole-genome sequences of the malaria parasite from around the world were analysed which allowed an overview of the worldwide variability, even though this analysis is biased by the availability of sequences, as the authors recognize.Despite the reduced sample size, the detailed analysis of haplotypes and identification of the location of breakpoints gives support to a single origin event for the 13-5++ parasites.The analysis of haplotype variation across the duplicated chromosome-11 segment identified breakpoints at varied locations that support multiple translocation events in natural populations. The authors suggest these translocations may be occurring at high frequency in meiosis in natural populations but are strongly selected against in most circumstances, which remains to be tested.Weaknesses:Reviewer comment: Relying on sequence data publicly available, that were collected based on diagnostic test positivity and that are limited by sequencing availability, limits the interpretation of the occurrence and relative frequency of the deletions.

However, we have uncovered more mechanisms than previously detected for hrp2 (involving MDR1) in SEA and South American parasites are likely detected by microscopy as RDTs were never introduced due to the presence of the deletions.

Reviewer comment: In the discussion, caution is needed when identifying the least common and most common mechanisms and their geographical associations. The identification of only one type of deletion pattern for Pfhrp2 may be related to these biases.

We added a section in the Discussion on the limitations of our study, which states the following, “Limitations of this study include the use of publicly available sequencing data that were collected often based on positive rapid diagnostic tests, which limits our interpretation of the occurrence and relative frequency of these deletions. This could introduce regional biases due to different diagnostic methods as well as limit the full range of deletion mechanisms, particularly pfhrp2.”

Reviewer comment: The specific objectives of the study are not stated clearly, and it is sometimes difficult to know which findings are new to this study. Is it the first study analyzing all the worldwide available sequences? Is it the first one to do long-read sequencing to span the entire duplicated region?

In the Introduction, we added, “The objectives of this study were to determine the pfhrp3 deletion patterns along with their geographical associations and sequence and assemble the chromosomes containing the deletions using long-read sequencing.”

We also added in the Discussion, “To the best of our knowledge, no prior studies have performed long-read sequencing to definitively span and assemble the entire segmental duplication involved in the deletions.”

Reviewer comment: Another aspect that should be explained in the introduction is that there was previous information about the association of the deletions to patterns found in chromosomes 5 and 11. In the short-read sequences results, it is not clear if these chromosomes were analysed because of the associations found in this study (and no associations were found to putative duplications or deletions in other chromosomes), or if they were specifically included in the analysis because of the previous information (and the other chromosomes were not analysed).

The former is correct. Chromosomes 5 and 11 were analyzed due to the associations found in this study, not from prior information. We have added the following sentence in the Results: “As a result of our short-read analysis demonstrating these three patterns and discordant reads between the chromosomes involved, chromosomes 5, 11, and 13 were further examined. No other chromosomes had associated discordant reads or changes in read coverage. ”

Reviewer comment: An interesting statement in the discussion is that existing pfhrp3 deletions in a low-transmission environment may provide a genetic background on which less frequent pfhrp2 deletion events can occur. Does it mean that the occurrence of pfhrp3 deletions would favor the pfhrp2 deletion events? How, and is there any evidence for that?

We should have stated more explicitly that selection would better be able to act on the now doubly deleted parasite versus a parasite with HRP3 still intact and weakly detectable by RDTs.Since fully RDT-negative parasites require a two-hit mechanism, where both *pfhrp2* and *pfhrp3* need to be deleted, and since there appear to be more mechanisms and drivers for *pfhrp3* deletions, this would create a population of parasites with one hit already and would only require the additional hit of *pfhrp2* deletion to occur to become RDT negative. So the point in the discussion being made is not that the *pfhrp3* deletion would favor *pfhrp2* deletion but rather that there is a population circulating with one hit already, which would make it more likely that the less frequent pfhrp2 deletion would result in a dual deleted parasite and therefore an RDT-negative parasite. The discussion has been modified to the following to try to make this point more clear. “In the setting of RDT use in a low-transmission environment, a pfhrp2 deletion occurring in the context of an existing pfhrp3 deletion may be more strongly selected for compared to pfhrp2 deletion occurring alone still detectable by RDTs.”

**Recommendations for the authors:**

**Reviewer #1 (Recommendations For The Authors):**
Reviewer comment: In the text, clonal propagation is the proposed hypothesis for the presence of near-identical copies of the chromosome 11 duplicated region. Even among the parasites showing variation between chromosomes, Figure 5 shows 3 haplotype groups with multiple sample members, which is also suggestive that these are highly related parasites. In addition to confirming COI status, it would be straightforward to calculate the genome-wide relatedness between/among parasites belonging to the same haplotype group. The assumption is that they are clones or highly related. A different finding would require more thought into potential genomic artifacts driving the pattern.

Thank you for this helpful suggestion. We confirmed the COI of each sample using THE REAL McCOIL. Six samples were not monoclonal, and we removed these samples from the downstream analysis to remove any contribution of polyclonal samples to the downstream haplotype analysis. Then, by using hmmIBD on whole-genome biallelic SNPs, we determined the whole-genome relatedness between the parasites. The haplotype groups do appear clonal though there appear to be several clonal groups within the larger groups of clusters 01 (n=28) and 03 (n=12) which combined with the variation seen within the 15.2kb region on chromosome 11/13, there appears to be different events that then lead to the same duplicated chromosome 11.

Reviewer comment: By way of validating the PathWeaver results, it could be useful to use another comparator method on the samples that are COI=1 or 2.

We have added an analysis based on biallelic SNPs to compare to the PathWeaver results, which produced similar results to help validate the PathWeaver results. We continued to use PathWeaver (Hathaway, in preparation), which is better able to detect variation relative to standard GATK4 analyses due to the refined local alignments from assembled haplotypes.

Questions regarding Methods:Reviewer comment: Were any metrics of genome quality factored into sample selection?

Yes, samples were removed if there was less than <5x median whole genome coverage. Additionally, several subsets of sWGA samples were removed based on visual inspection. These details have been added to the methods section.

Reviewer comment: How were polyclonal samples treated to ensure they did not produce analysis artifacts?

The read-depth analysis required zero coverage across the regions of *pfhrp2/pfhrp3,* which made it so that most of the samples analyzed were monoclonal (or polyclonal infections of only deleted strains). We have now used THE REAL McCOIL on whole genome SNPs to determine COIs. Six samples were identified as polyclonal, and we removed them for the analysis and updated the manuscript. Their removal did not significantly impact the results or conclusions.

Reviewer comment: How was local realignment of short-read data performed? Was this step informed by the conserved, non-paralogous genomic regions, or were these only used for downstream variant analysis?

No local realignment of short-read data was performed. The analysis was either read depth or de novo assembly from reads from specific regions. Regarding the de novo assembly, variant calls were replaced by complete local haplotypes, and a region was typed based on the haplotype called for the region.

Reviewer comment: For read-depth estimation, what cutoffs were used to classify windows as deletion, WT, or duplication? How much variability was present in the data? The plot legends imply a continuous scale, but in reality, only 3 discrete colors are used (0, 1, 2), so these must represent the data after rounding.

These have been added to the manuscript. See response to Reviewer #1 questions #2 and #3 above

Reviewer comment: Similarly, what thresholds were used for mapping the long-reads? In Fig S21, it appears there is a high proportion of discordant reads.

Long reads were mapped using minimap2 with default settings. For Figure 21, since it is from the mappings to 3D7 chromosome 11 and hybrid 3D7 13-11 chromosome, the genome from the duplicated region from the blue bar underneath is identical, so reads are expected to map to both since the genome regions are identical. The significance of this figure and Figure 4 is the number of long reads that span the whole chr11/13 duplicated region connection the 3D7 chromosome 11 and the hybrid proving that there are reads that start with chromosome 13 sequence and end with chromosome 11 sequence and the lack of reads that span from chromosome 13 into the 3D7 chromosome 13.

Reviewer comment: The section on the mdr1 breakpoints is too vague.

We have updated the methods section to be more explicit about how these breakpoints were determined.

Reviewer comment: I assume that the "Homologous Genomic Structure" section of the Methods is the number analysis that was alluded to in the Results? As with other sections, this needs more information on exact methods and tools

We have now updated the methods section to include exactly how the nucmer commands were run.

Smaller comments:Reviewer comment: Introduction sub-header: "Precise *pfhrp2* and..."

We have corrected the sub-header.

Reviewer comment: Results (p.5) cite Table S4 instead of S3

We have corrected this to Table S3.

Reviewer comment: Results (p.5) "We identified 27 parasites with pfhrp2 deletion, 172 with pfhrp3 deletion, and 21 with both pfhrp2 and pfhrp3 deletions." This sentence makes it sound like they are 3 mutually exclusive categories. I'd suggest a rewording like "We identified 27 parasites with pfhrp2 deletion and 172 with pfhrp3 deletion. Of these, 21 contained both deletions."

We have re-worded this sentence to the following: “We identified 26 parasites with pfhrp2 deletion and 168 with pfhrp3 deletion. Twenty field samples contained both deletions; 11 were found in Ethiopia, 6 in Peru, and 3 in Brazil, and all had the 13-11++ pfhrp3 deletion pattern.”

Reviewer comment: The annotations used for the deletions differ between the text and the figures. It would be easier for the reader to harmonize the two if these matched.

The figures have been updated to reflect the annotations of the text.

Reviewer comment: Figure numbering does not match the order they are first referenced in the text

The figure numbers have been updated to match the order in which they are first referenced.

Reviewer comment: Results (p. 8) there is no Table S4

This has been changed to Table S3.

Reviewer comment: Results (p.8) mention a genome-wide number analysis, but I couldn't find these results. The referenced figure is for the duplicated region only.

We have updated to point to the correct location of the nucmer results by adding a supplemental table with the results and updated to point to the correct figure.

Reviewer comment: Discussion typo: "Here, we used publicly available short-read and long-read *short-read sequencing data* from..."

This was not a typo, as we used publicly available PacBio long-read data and then generated new Nanopore long-read data. However, we did clarify this in the sentence.

**Reviewer #2 (Recommendations For The Authors):**
IntroductionReviewer comment: "(...) suggesting the genes have important infections in normal infections and their loss is selected against". The word "infections" is in place of "role", etc.

We have changed the word accordingly.

ResultsReviewer comment: In the section "Pfhrp2 and pfhrp3 deletions in the global *P. falciparum* genomic dataset" it is mentioned the number of parasites with each deletion and where it is more common. "We identified 27 parasites with pfhrp2 deletion, 172 with pfhrp3 deletion, and 21 with both pfhrp2 and pfhrp3 deletions." and "Across all regions, pfhrp3 deletions were more common than pfhrp2 deletions; specifically, pfhrp3 deletions and pfhrp2 deletions were present in Africa in 43 and 12, Asia in 53 and 4, and South America in 76 and 11 parasites." It is not clear where the 21 parasites with both pfhrp2 and pfhrp3 deletions are located.

We have specified the following in the Results section: “We identified 26 parasites with pfhrp2 deletion and 168 with pfhrp3 deletion. Twenty field samples contained both deletions; 11 were found in Ethiopia, 6 in Peru, and 3 in Brazil, and all had the 13-11++ pfhrp3 deletion pattern”

Reviewer comment: "It should be noted that these numbers are not accurate measures of prevalence given that most WGS specimens have been collected based on RDT positivity." This, combined with the fact that subtelomeric regions are difficult to sequence and assembly, means these numbers are underestimated. I believe it should be more stressed in the text.

We have added the following sentence, “*Furthermore, subtelomeric regions are difficult to* sequence and assemble, meaning these numbers may be significantly underestimated.”

Reviewer comment: In the section "Pattern 13-11++ breakpoint occurs in a segmental duplication of ribosomal genes on chromosomes 11 and 13", Figures 2a and 2b should be mentioned in the text instead of just Figure 2.

We have specified Figures 2a and 2b in the text now.

Figures and Tables:Reviewer comment: Figure 2: I believe the color scale for percentage of identity is unnecessary given that the goal is to show that the paralogs are highly similar, and not that there is a significant difference between 0.99 and 0.998.

Updated the color scale to represent the number of variants between segments rather than percent identity which ranges between 55-133 so that it represents something more discreet than 0.99 and 0.998.

Reviewer comment: Adjust Figure 2b and the size of supplementary figure legends.Supplementary Figure 5-15: the legends are hard to read.

All legends have been adjusted to be much more readable.

**Reviewer #3 (Recommendations For The Authors):**
Some minor suggestions:Reviewer comment: The order of the figures should follow the flow of the text, for example, Figure 5 appears in the text between Figure 1 and Figure 2.

We have reordered the figures according to the order in which they appear in the text.

Reviewer comment: Page 3 - "deleted parasites" - better to use: pfhrp2/3-deleted parasites.

We have edited this accordingly.

Reviewer comment: Define the acronyms the first time they are used, e.g. SEA.

We have defined the acronyms accordingly.

Reviewer comment: In the figures where pfmdr1 appears, indicate the correspondence to the full name of the gene that appears in the legend (multidrug resistance protein 1).

Legends updated.

Reviewer comment: Page 5 - Table S4 is missing.

We apologize for our typo. There is no Table S4. We meant to refer to Table S3, which has been updated accordingly.

Reviewer comment: Page 5 - "We identified 27 parasites with pfhrp2 deletion, 172 with pfhrp3 deletion, and 21 with both pfhrp2 and pfhrp3 deletions" - is it "and 21..." OR "from which, 21..."?

We have reworded the sentence to the following: “We identified 26 parasites with pfhrp2 deletion and 168 with pfhrp3 deletion. Twenty field samples contained both deletions; 11 were found in Ethiopia, 6 in Peru, and 3 in Brazil, and all had the 13-11++ pfhrp3 deletion pattern.”

Reviewer comment: Page 5 - "most WGS specimens have been collected based on RDT positivity." - explain better which tests are done - to detect pfhrp2, pfhrp3 or both?Co-occurrence is not detected?

We used all publicly available WGS data that spanned over 30 studies, and the exact details of what RDTs were used are not readily available to fully answer this question. Though the exact details of RDTs are not known, this does not affect the deletion patterns found in the genomic data but does limit the ability to comment on how this affects prevalence. We have updated the manuscript to the following to be more explicit that we don’t have the full details: “It should be noted that these numbers are not accurate measures of prevalence, given that the publicly available WGS specimens utilized in this analysis come from locations and time periods that commonly used RDT positivity for collection”

Reviewer comment: Supplementary Figure 1 - Legend for "Pattern" - what is the white?

The “Pattern” refers to pfhrp3 deletion pattern with “white” being no pfhrp3 deletion. The annotation title has been changed to “pfhrp3- Pattern” to make this more clear and added to the text of the legend the following:”Of the 6 parasites without HRP3 deletion (marked as white in pfhrp3- Pattern column for having no pfhrp3 deletion),...”

Reviewer comment: Supplementary Figure 8 - explain the haplotype rank. How was it obtained?

The haplotype rank is based on the prevalence of the haplotype. To clarify this better the following has been added to the caption “Each column contains the haplotypes for that genomic region colored by the haplotype prevalence rank more prevalent have a lower rank number, with most prevalent having rank (1) at that window/column. Colors are by frequency rank of the haplotypes most prevalent haplotypes have rank 1 and colored red, 2nd most prevalent haplotypes are rank 2 and colored orange, and so forth. Shared colors between columns do not mean they are the same haplotype. If the column is black, there is no variation at that genomic window.”

Reviewer comment: Figure 1 - Pattern in legend appears 11++13- but in text it is always referenced as 13-11++

Figure legend has been updated to reflect the annotation within the text

Reviewer comment: Page 6 - pattern 13- is which one(s) in Figure 1?

This refers to the 13- with TARE1 sequence detected, the text has been updated to “*(pattern 13-TARE1)*” and the legend of Figure 1 has been updated so these statements match more closely.

Reviewer comment: Page 7 - states "The 21 parasites with pattern 13-" and refers to Supplementary Figure 3 which presents "50 parasites with deletion pattern 13-". I believe this is pattern 13- unassociated with other rearrangements but it should be made clear in the text and legend of the supplementary figure.

Thank you, you are correct. The manuscript has been updated in two locations for better clarity. The text has been updated to be “The 20 parasites with pattern 13-TARE1 without associated other chromosome rearrangements had deletions of the core genome averaging 19kb (range: 11-31kb). Of these 13-TARE1 deletions, 19 out of 20 had detectable TARE1 (pattern 13-TARE) adjacent to the breakpoint, consistent with telomere healing.” The Supplemental Figure 3 legend has been updated to “for the 48 parasites with pfhrp3 deletions not associated with pattern 13-11++”

Reviewer comment: Supplementary figure 25 - "regions containing the pfhrp genes (lighter blue bars below chromosomes 11 and 13)" - the light blue bars are shown below chromosome 8 and 13; what is the difference between yellow and pink bars (telomere associates repetitive elements in the truncated legend)?

The yellow bars are associated with the telomere-associated repetitive element 3 and the pink bars are telomere-associated repetitive element 1. To add clarity the legend has been updated to be “The yellow (TARE3) and pink (TARE1) bars on the bottom of the chromosomes represent the telomere-associated repetitive elements found at the end of chromosomes.”

Reviewer comment: It would be helpful to have a positioning scale in the figures.

Most plots have y-axis and x-axis with the genomic positioning labeled which can serve as a positioning scale so we opted not to add more to the figures to keep them less crowded. Other plots have regions plotted in genomic order but are all relatively positioned which prevents the usage of a positioning scale, we tried to clarify this by adding more details to the captions of these figures.

Reviewer comment: Legend of Figure 6 - The last paragraph seems to be out of place

We have deleted the last sentence in the legend of Figure 6 accordingly.